# Real-Time Efficient FPGA Implementation of the Multi-Scale Lucas-Kanade and Horn-Schunck Optical Flow Algorithms for a 4K Video Stream

**DOI:** 10.3390/s22135017

**Published:** 2022-07-03

**Authors:** Krzysztof Blachut, Tomasz Kryjak

**Affiliations:** Embedded Vision Systems Group, Computer Vision Laboratory, Department of Automatic Control and Robotics, AGH University of Science and Technology, Al. Mickiewicza 30, 30-059 Krakow, Poland

**Keywords:** optical flow, multi-scale, 4K resolution, FPGA, real-time processing, vision system, Lucas–Kanade algorithm, Horn–Schunck algorithm

## Abstract

The information about optical flow, i.e., the movement of pixels between two consecutive images from a video sequence, is used in many vision systems, both classical and those based on deep neural networks. In some robotic applications, e.g., in autonomous vehicles, it is necessary to calculate the flow in real time. This represents a challenging task, especially for high-resolution video streams. In this work, two gradient-based algorithms—Lucas–Kanade and Horn–Schunck—were implemented on a ZCU 104 platform with Xilinx Zynq UltraScale+ MPSoC FPGA. A vector data format was used to enable flow calculation for a 4K (Ultra HD, 3840 × 2160 pixels) video stream at 60 fps. In order to detect larger pixel displacements, a multi-scale approach was used in both algorithms. Depending on the scale, the calculations were performed for different data formats, allowing for more efficient processing by reducing resource utilisation. The presented solution allows real-time optical flow determination in multiple scales for a 4K resolution with estimated energy consumption below 6 W. The algorithms realised in this work can be a component of a larger vision system in advanced surveillance systems or autonomous vehicles.

## 1. Introduction

Motion detection is one of the most important elements in the field of image processing and analysis. This motion can result from a change in the position of an object, a camera or both of them. For a human, this task is easy and natural, even when comparing just two images, as in Figure 1. In addition, this is due to our evolutionary adaptation, as movement usually means a potential threat that needs to be quickly analysed and reacted to.

In the simplest case, the motion can be detected by subtracting two subsequent frames from a video sequence. This allows to see where the movement (change) has occurred, but does not provide significant information about its direction and speed. However, this information is essential in more advanced vision systems. Hence, the concept of optical flow was introduced, i.e., a vector field describing the movement of a pixel between two images from a sequence, in which two values are associated with each pixel—its horizontal and vertical displacements. The ratio of these two values makes it possible to determine the direction of the movement, while their magnitude allows to determine the speed.

Algorithms that perform this type of task are increasingly used in everyday life. They are very popular in autonomous vehicles for detecting and tracking pedestrians or other vehicles. They are also widely used in aerial platforms for obstacle detection, stabilisation, and navigation. Other areas, in which optical flow determination algorithms are used, include monitoring and surveillance systems, gesture recognition and control, 3D scene reconstruction, self-displacement estimation, object segmentation, and depth estimation.

In the last 40 years, many different methods have been proposed to determine optical flow. The first group of methods is based on gradient calculations, analysing the spatial and temporal movements of pixels and then finding a minimum of certain functions. Examples of such algorithms include the work of Horn and Schunck (HS) [2] or Lucas and Kanade (LK) [3]. They are simple yet effective and relatively easy to parallelise, thus they can operate in real time. These methods are presented in more detail in Section 2. The research on more efficient ways to determine the flow has led, over time, to the use of variational methods, such as TV-L1 [4], which is based on the HS algorithm.

The second group consists of block methods, sometimes called correlation methods, in which the operation scheme is based on determining the maximum correlation of a certain area between two frames. The displacement of this area is equivalent to the determined optical flow. The most widely used metric in this task is SSD (sum of squared differences) in a small neighbourhood (context) *W*. In this type of methods, as the calculations in bigger neighbourhoods are very costly, many scales are often used, e.g., in the work of Anandan [5]. In addition, a common approach is to restrict the analysis to predetermined characteristic points (e.g., corners). This results in a sparse flow, but the results are less prone to errors. However, to obtain dense flow, such as those in gradient-based algorithms, an additional operation (potentially a complex interpolation) is needed.

The third group contains the approaches based on phase properties and analysis in the frequency domain, which was first proposed in the work of Fleet and Jepson [6]. Noticeable differences in results can be caused by, for example, changes in illumination, to which phase methods are much more robust. These methods use Gabor filters and the Fourier transform. However, the biggest disadvantage of the phase-based approach is its high computational complexity and the need to use hardware platforms that allow parallel computations. Therefore, there are relatively few publications dedicated to them in the literature.

More popular, especially in recent years, is another group of methods. The main limitation of gradient-based algorithms is the assumption of a small pixel displacement. The proposed new approach has enabled the detection of larger ones, as it is based on the detection and matching of features points and then on the minimisation of a certain function. Therefore, solutions such as SIFT Flow [7] have been proposed, where a well-known SIFT (scale-invariant feature transform) algorithm is used to detect and match feature points and later calculate optical flow, regardless of the displacement between frames. To obtain dense flow, an energy function is used in which assumptions of a constant pixel brightness and smoothness of the flow are included. Another solution of this type is LDOF (large displacement optical flow) [8], in which the authors match specific regions determined by a segmentation algorithm using multiple scales. Interestingly, the matching uses the Horn–Schunck method, as it can find a better candidate than the nearest neighbour method. To obtain a dense flow from a sparse one, a variational approach is used, in which multiple elements are taken into account: pixel brightness, gradient, matched regions, and detected edges. However, these methods are computationally demanding and not suitable for real-time operation.

In recent years, neural networks have become very popular in many tasks, with particular emphasis on convolutional networks and deep learning. Therefore, a new group of algorithms has also emerged for the optical flow determination task, which is based on deep convolutional neural networks (DCNNs).

First of the notable solutions was FlowNet [9], where two architectures were used: FlowNetSimple, in which two images were fed to the input of the network as one with a larger third dimension, and FlowNetCorr with a correlation layer, where the merging of features from two images took place in a small neighbourhood. A continuation of this work resulted in the publication of FlowNet 2.0 [10] and FlowNet 3.0, which better handle disruptions and small displacements, ensures smoother flow and better preserved edges. Another solution is SPyNet [11], which uses a pyramid of images and an individual network (with only 5 layers) for each of 5 scales, with the networks trained independently. A certain development of SPyNet is Deeper SPyNet [12] with several improvements to the network structure to reduce the dimensions of the convolutional masks and to add more layers.

Currently, one of the best networks for optical flow is LiteFlowNet [13], which uses a multi-scale approach. However, in this case, instead of generating a pyramid of images, a pyramid of features was generated. Thus, in the warping stage, modifications were performed in the feature space rather than by moving pixels in the image. The application also uses regularisation to reduce the influence of noise and smooth the flow, while conserving the edges. The continuation of the work resulted in LiteFlowNet2 [14] and LiteFlowNet3. Compared to their predecessor, these networks yield higher accuracy flow in shorter time, as the authors made some changes to the training process and the architecture of the network. A solution with a comparable accuracy and size to LiteFlowNet2 is PWC-Net [15] and its successor PWC-Net+. The authors used a similar approach—in the multi-scale method, a feature pyramid is constructed instead of an image pyramid. The whole structure of the method is quite complex, as it uses different networks for feature extraction, optical flow determination, and post-processing of the flow.

In summary, in the optical flow determination task (as in many others), DCNN-based solutions have become state of the art in terms of accuracy. The trend in new solutions is not only to increase the accuracy, but also to reduce the number of parameters of the network and the processing time. However, they cannot process very high-resolution video streams in real time due to their complexity and number of parameters—the best solutions (such as LiteFlowNet2) can process up to 25–30 frames per second on Nvidia 1080 GTX GPU for the images of size 1024 × 436 pixels, which is nearly 20× smaller than a frame in 4K resolution. They also require a huge amount of training data and do not have as strong of a theoretical background as the “classical” methods do.

Currently, the main challenges in the field of optical flow are the occurrence of large displacements and overlaps, accurate detection of object edges, removing the artefacts, correct operation in different weather conditions, and processing high-resolution data in real time. One of the mentioned problems—detection of large displacements—can be solved by using the multi-scale method. The reason for its use is due to the assumptions present in the “traditional” methods, which only allow for the correct determination of small displacements. More details about this method are presented in Section 2.3.

Many solutions published have been realised on a general purpose processor. However, several complex methods are too computationally demanding to satisfy the requirement of working in real-time, especially for high-resolution video streams. Currently available cameras allow high-resolution image acquisition—from the HD standard (1280 × 720 pixels) through Full HD (1920 × 1080 pixels) to 4K (3840 × 2160 pixels). This improves the accuracy of the analysis, but at the same time presents a major computational challenge. Interesting alternatives can be found in the literature, such as ref. [16] in which the authors calculate the flow using a neural network for 4K images reduced 4× in each dimension and then reconstruct it in the original resolution—however, with this approach, there is a risk of losing some details. With the increasing popularity of embedded systems that enable the acceleration of specific calculations, multiple vision-based algorithms have been implemented on them, presenting real-time operation with low energy consumption. Therefore, optical flow algorithms can also be implemented on devices operating in a parallel manner, such as graphics processing units (GPUs) or field programmable gate arrays (FPGAs). In both cases, numerous solutions have already been published—for the former, mainly DCNN-based, which are outside the scope of this work, while for the latter, a comprehensive review is presented in Section 3.

The main contributions of this paper are as follows:Proposition of an architecture able to process Horn–Schunck and Lucas–Kanade optical flow computation algorithms in multi-scale versions in real-time in Ultra HD (4K) resolution on an FPGA platform, which, to our best knowledge, has not been done before.Efficient implementation of the multi-scale method, taking advantage of processing different number of pixels simultaneously depending on the scale and without using additional external memory to store temporal values.

The structure of the paper is as follows. Section 2 presents theoretical aspects of the optical flow determination by the HS and LK algorithms along with the multi-scale method. Section 3 is devoted to the publications in which optical flow algorithms were implemented on FPGA platforms. In Section 4, the proposed hardware implementation of the OF algorithms is presented in details. The evaluation of the proposed solution is presented in Section 5, while the last Section 6 contains a summary of the implementation, along with ideas for future work.

## 2. The Horn-Schunck and Lucas-Kanade OF Computation Algorithms

To obtain correct optical flow values using gradient-based optical flow algorithms, certain assumptions have to be met. The first is the small displacement of the object between two consecutive frames, which is usually met for objects that move slowly. In other cases, additional modifications are necessary—e.g., increasing the frame rate (fps) in the image acquisition device or using a multi-scale method. The second assumption is the constant pixel brightness (in greyscale) between consecutive image frames. For two frames captured at moments in time, denoted as *t* and *t* + 1, the equation associated with the pixel brightness for the image *I* can be written as Equation (Equation 1) and differentiated with respect to time, as in Equation (Equation 2).
(1)I(x(t),y(t),t)=I(x(t+1),y(t+1),t+1)=const
(2)$$\frac{dI(x(t),y(t),t)}{dt}=0$$
where *x* and *y* correspond to the horizontal and vertical location of the pixel in the image.

When the expression is expanded into Taylor series, an approximation can be made by omitting derivatives of orders higher than the first. This yields the initial Equation (Equation 3) in gradient optical flow methods.
(3)$$\frac{\partial I}{\partial x}\frac{\partial x}{\partial t}+\frac{\partial I}{\partial y}\frac{\partial y}{\partial t}+\frac{\partial I}{\partial t}=0$$
where:$\frac{\partial I}{\partial x},\frac{\partial I}{\partial y}$—spatial derivatives, $\frac{\partial I}{\partial t}$—temporal derivative, $\frac{\partial x}{\partial t},\frac{\partial y}{\partial t}$—optical flow.

This equation can be written in a more compact form (Equation (Equation 4)), using the pixel offset defined as (*u, v*).
(4)$$I_{x}u+I_{y}v+I_{t}=0$$

In a general case, it is impossible to obtain an unambiguous solution of the equation with two unknowns (*u, v*). For this reason, several methods have been proposed in the literature to solve this problem—both global and local ones. In the global methods, the features of the entire image are taken into account, while in the local methods, the value of the determined flow is obtained only on the basis of the neighbouring pixels (usually by calculating their average). In this work, both approaches are presented in the subsections below and then implemented on a hardware platform—the HS algorithm (global method) and the LK algorithm (local method).

### 2.1. Horn–Schunck Algorithm

In the work of Horn and Schunck [2], the authors proposed an additional assumption to solve Equation (Equation 4) that the optical flow should be smooth over the entire image. Therefore, for each pixel, the calculated flow is similar in a small neighbourhood. Such a problem formulation made it possible to reduce it to the minimisation of a function denoted as Equation (Equation 5). In the expression, two parts can be distinguished. One is responsible for the constant brightness of a pixel (like in Equation (Equation 4)). The other is responsible for the regularisation of the flow across the image, expressed as the sum of the squared magnitudes of the spatial gradients calculated on flow components *u* and *v*. Its effect can be controlled by changing the constant numeric value of the parameter α.
(5)$$E=\;\int \int \;[{\left(I_{x}u+I_{y}v+I_{t}\right)}^{2}+\alpha^{2}\left({\left||\nabla u|\right|}^{2}+{\left||\nabla v|\right|}^{2}\right)]dxdy$$

The Horn–Schunck (HS) algorithm is a global method for determining the optical flow. Therefore, a global minimum of the function is searched for, whose increasingly accurate approximations can be obtained in an iterative manner using the formulas Equations (Equation 6) and (Equation 7).
(6)$$u_{n+1}=\bar{u_{n}}-\frac{I_{x}\left(I_{x}\bar{u_{n}}+I_{y}\bar{v_{n}}+I_{t}\right)}{\alpha^{2}+I_{x}^{2}+I_{y}^{2}}$$
(7)$$v_{n+1}=\bar{v_{n}}-\frac{I_{y}\left(I_{x}\bar{u_{n}}+I_{y}\bar{v_{n}}+I_{t}\right)}{\alpha^{2}+I_{x}^{2}+I_{y}^{2}}$$
where:$\bar{u_{n}},\bar{v_{n}}$—average velocity in the neighbourhood.

### 2.2. Lucas–Kanade Algorithm

Lucas and Kanade [3] proposed a different approach to solve the Equation (Equation 4). Instead of searching for the global minimum of a function, the authors considered a small neighbourhood of each pixel. This idea is based on the observation that a pixel moves in the same way as its nearest neighbours. Therefore, the assumption introduced in their algorithm (LK) only needs to be satisfied locally, contrary to the HS algorithm. For *n* pixels from the surrounding area, denoted as *p_1_*, *p_2_*, …, *p_n_*, the equation related to the pixel brightness can be written in matrix form as Equation (Equation 8).
(8)$$\left(\begin{array}{cc} I_{x}\left(p_{1}\right) & I_{y}\left(p_{1}\right)\\ I_{x}\left(p_{2}\right) & I_{y}\left(p_{2}\right)\\ \vdots & \vdots \\ I_{x}\left(p_{n}\right) & I_{y}\left(p_{n}\right) \end{array}\right)\left(\begin{array}{c} u\\ v \end{array}\right)=-\left(\begin{array}{c} I_{t}\left(p_{1}\right)\\ I_{t}\left(p_{2}\right)\\ \vdots \\ I_{t}\left(p_{n}\right) \end{array}\right)$$

To simplify the notation of subsequent calculations, the following symbols can be introduced in Equation (Equation 9).
(9)$$A=\left(\begin{array}{cc} I_{x}\left(p_{1}\right) & I_{y}\left(p_{1}\right)\\ I_{x}\left(p_{2}\right) & I_{y}\left(p_{2}\right)\\ \vdots & \vdots \\ I_{x}\left(p_{n}\right) & I_{y}\left(p_{n}\right) \end{array}\right),\;d=\left(\begin{array}{c} u\\ v \end{array}\right),\;b=-\left(\begin{array}{c} I_{t}\left(p_{1}\right)\\ I_{t}\left(p_{2}\right)\\ \vdots \\ I_{t}\left(p_{n}\right) \end{array}\right)$$

In the expression obtained in this way, there are more equations than unknowns, thus the problem posed can be reduced to the minimisation of the expression of Equation (Equation 10), as proposed by the authors of the algorithm.
(10)$$Ad=b⟶{\min∥Ad-b∥}^{2}$$

To determine the solution of *d*, the equation was multiplied by *A^T^W* on both sides, which led to Equation (Equation 11). The weight matrix *W* takes into account a different impact of the pixels depending on their distance from the analysed pixel. In the simplest case, *W* can be a unitary matrix, while using a Gaussian-like mask increases the weights of the closest pixels.
(11)$$\left(A^{T}WA\right)d=A^{T}Wb$$

This equation can also be written in equivalent form (Equation (Equation 12)), in which the products of individual derivatives are summed together with the corresponding weights *w_i_*.
(12)$$\left(\begin{array}{cc} \sum w_{i}I_{x}I_{x} & \sum w_{i}I_{x}I_{y}\\ \sum w_{i}I_{x}I_{y} & \sum w_{i}I_{y}I_{y} \end{array}\right)\left(\begin{array}{c} u\\ v \end{array}\right)=-\left(\begin{array}{c} \sum w_{i}I_{x}I_{t}\\ \sum w_{i}I_{y}I_{t} \end{array}\right)$$

The existence of a solution to this equation depends on the invertibility of the matrix *A^T^WA*. In other words, if its determinant is different from 0, there is an optical flow expressed by Equation (Equation 13).
(13)$$\left(\begin{array}{c} u\\ v \end{array}\right)={\left(A^{T}WA\right)}^{-1}A^{T}Wb$$

In reality, the invertibility of the matrix *A^T^WA* does not always guarantee a correct solution. For this reason, additional conditions are proposed in the literature in which the eigenvalues of this matrix are taken into account—they cannot be too small, and their quotient cannot be too large. Determining whether the eigenvalues of a matrix satisfy these conditions is usually done by a comparison with a threshold.

### 2.3. Multi-Scale Method

In many practical applications, the presented optical flow calculation algorithms are not sufficient, mainly due to the presence of fast moving objects or insufficiently high sampling rate of the acquisition device. In such situations, the multi-scale approach is most widely used. It differs from the standard version (i.e., single scale) in the usage of the same image in multiple resolutions during the calculations.

First, an image pyramid is constructed for each of the input images that are used to calculate the optical flow as in [5]. The frame is downscaled multiple times (usually no more than 3, which means 4 scales in total) and the dimensions of the image are usually reduced twice between the scales. Next, the optical flow is determined for the pair of images in the smallest scale (“top of the pyramid”) according to the selected algorithm (HS, LK or other). The resulting flow is then upscaled to the size of a larger scale—both the dimensions of the vector field and its values are increased (usually twice). Then, the previous frame is modified according to the optical flow by shifting the pixels to calculated places (this is often referred to as warping). The purpose of such a procedure is to make a motion compensation, so in larger scales, the assumption of a small pixel displacement is not violated. Then, using a modified previous frame and a downscaled current frame, the optical flow is calculated again, but this time in a larger scale. The entire computation scheme is repeated until the flow is calculated for a pair of images in the initial size.

Figure 2 shows a schematic computation procedure of the optical flow using the multi-scale method using an exemplary sequence *Camera motion* from the MIT CSAIL dataset [1]. It can be concluded that in smaller scales, the optical flow gives rough information about the overall object displacement, while in larger scales, the accuracy of the flow is increased.

## 3. FPGA Implementations of Optical Flow Methods

Because of the considerable complexity of the hardware implementation of algorithms on an FPGA platform, most published applications for optical flow determination have been realised using gradient Lucas–Kanade and Horn–Schunck methods, which are relatively simple. On the other hand, many complex algorithms are not suitable for hardware implementation in a pipelined system because of the operations performed or insufficient amount of available computing resources. However, apart from implementations of gradient methods, several papers can be found in the literature where other approaches have been used to determine optical flow on an FPGA platform.

In the work [17], the optical flow calculation was based on tensors that determine the orientation in the spatio-temporal domain. Using navigation as an exemplary task, the authors noted that the speed of obtaining results is more important than their accuracy and achieved a processing of 640 × 480 @ 64 fps video stream. The computations performed on the tensors are very similar to those in the Lucas–Kanade algorithm. The authors used temporal smoothing over 5 frames and a LUT (Look-Up Table)-based divider to reduce resource utilisation.

A similar approach to the previous work was used in [18], where the authors compared implementations on two platforms commonly used in vision-based tasks: FPGA and GPU. As a result, a processing of 640 × 480 @ 150 fps was realised for GPU, while for FPGA, at 64 fps. However, according to the authors’ calculations, with applying some optimisations, using a better platform and a memory controller, the frame rate could be increased to 300. Perhaps the most interesting conclusion is the similarity of the results with 12× longer implementation time on FPGA, which requires additional skills and experience for efficient logic design. However, compared to GPUs, FPGAs are superior in several other aspects, which include the ability to operate without a computer as a host, smaller size and power consumption, and higher flexibility in programming.

Among the hardware implementations of optical flow calculation on an FPGA platform, block methods have also been used. In one of such works [19], a processing of 640 × 480 @ 39 fps video stream was realised using SAD (sum of absolute differences) matching. An image pyramid of three levels was used in the algorithm to enable the detection of larger displacements. In the algorithm, constant pixel brightness was assumed as well as a similar movement of the blocks belonging to the same object, which led to the minimisation of an energy function to eliminate irregular matches and smooth the flow. The results obtained in the work were better than in classical gradient methods, but the quality of the matching depends on many factors, including block size, search region, and starting point.

### 3.1. Lucas-Kanade FPGA Implementations

Several implementations of the LK algorithm on FPGA platform have already been published in the literature. In the vast majority of them, the authors focused on a single-scale version, using similar components: low-pass filtering, derivatives calculation, summation and matrix inversion. In addition, in most cases, an external memory controller was also used, which was necessary to store and read the previous frame. Sometimes temporal smoothing over multiple frames was used to reduce the impact of disturbances occurring on single frames, but at the cost of increased memory utilisation.

In one of the first works [20] to address the implementation of the LK algorithm, an architecture that allows the processing of 320 × 240 @ 30 fps video stream was described. A floating point version of the matrix inversion module was used to reduce resource utilisation.

Further work on the LK algorithm resulted in significant improvements in the system throughput. In the publication [21], an architecture was presented that allowed the processing of 800 × 600 @ 170 fps video stream. Thanks to the usage of high frame rate cameras that allowed for more frequent image acquisition, the transitions between successive frames were smoother and resulted in a reduction of used resources.

In a subsequent study by the same authors [22], the system realised allowed the processing of 640 × 480 @ 270 fps video stream. Several versions of the LK algorithm were presented, including ones that use a floating-point or a fixed-point module for the matrix inversion. In addition, a version of the algorithm that returns approximate results at the cost of a very small resource utilisation was also presented.

A partially modified approach to the Lucas–Kanade algorithm was used in the work of [23]. The proposed system allowed processing of 1200 × 680 @ 500–700 fps video stream. This approach differed from the traditional one, as the calculations were performed for groups of several pixels instead of individual ones. In addition to this, a verification of the results was applied—values smaller than a set threshold were zeroed.

Reducing the use of external memory was the motivation of the work [24], which was able to process 800 × 600 @ 196 fps video stream. The idea of reduced external memory usage was based on the fact that only 25% of pixels were stored for each frame. In the case of the first image, pixels with odd indexes were saved into memory; for the second, pixels with even indexes; for the third, again with odd indexes, and so on. Based on the pixels from different frames, an image was reconstructed, which then fed up the module responsible for calculating the derivatives. The image fragments stored in memory were the results of Gaussian blurring, so the differences between pixels on successive frames were small and allowed correct results to be obtained.

In [25], the Vivado HLS tool was used for the hardware implementation of the Lucas–Kanade algorithm for the processing of 1920 × 1080 (Full HD) @ 123 fps video stream. In this work, a different scheme for summing elements in the window was suggested. Instead of repeatedly calculating the sum of elements in columns, these values were stored in additional registers, adding to them elements from the right-hand side column of the sliding window and subtracting elements from the left-hand side column. This allowed fast and effective summations in windows of size up to 53 × 53 pixels (px), making the results more consistent for moving objects. Several optimisations of resource utilisation were also proposed—specifically for Block RAM and DSP multipliers.

To determine the motion of fast-moving objects, the number of image frames captured per second can be increased if the camera allows that. Increasing the image acquisition rate reduces the pixel displacement between two consecutive frames. If this approach is not feasible, a multi-scale method can be used in the algorithm. However, due to its complexity and the necessity of a pyramid image generation, such hardware implementations are rare.

In work [22], a multi-scale version of the Lucas–Kanade algorithm was realised for 640 × 480 @ 32 fps video stream, fulfilling the requirement for real-time processing. In addition to the modules used in the single-scale version, several additional components were added. To generate a pyramid of images, an approach based on selecting every second pixel in every second row was used, while in order to upscale the image twice, bilinear interpolation was used. A separate module was dedicated to image warping and motion compensation. The results obtained were median-filtered to remove single incorrect values and then summed from different scales to visualise the final result.

In the literature, there are examples of simplified approaches to dividing operation during matrix inversion in the LK algorithm. The authors of the paper [26] assumed that since the pixel displacement is small, the calculated flows would have small values—specifically, between −5 and +5 pixels. Thus, it was possible to limit the output values of the divider, and the operation itself was reduced to bit shifts and cascades of comparators. This limitation of the flow values forced the authors to use a multi-scale method with three pyramid levels, which was controlled by a state machine. Unfortunately, neither the resolution nor the frequency of the processed data was included in the paper.

Our previous work [27] of the multi-scale LK algorithm on FPGA was able to process 1280 × 720 (HD) @ 50 fps video stream. Typical modules used in the LK algorithm were implemented—conversion to greyscale, Gaussian blur, derivatives calculation, summations in a window 5 × 5 px, matrix inversion and thresholding of the results. The modules used in the multi-scale version (downscaling, upscaling, warping and flows summation) were implemented similarly to those found in the work [22]. The lessons learnt and ideas for possible improvements were used to implement the LK algorithm in this work.

Some works focus on specified optical flow processors, realised as VLSI architectures. In one of these works [28] a multi-scale version of the LK algorithm was implemented firstly on FPGA, then on designed VLSI processor. On the latter, the authors achieved processing of 640 × 480 (VGA) @ 30 fps video stream with three scales, using only 600 mW of energy. The authors used a filter that combined information from the next frame with the current frame instead of using them separately. The proposed architecture was also able to process four pixels simultaneously, using four processors.

Another work is [29], in which the authors proposed adaptive multi-scale processor able to process the LK algorithm with different density, precision, energy consumption and number of scales (up to 4). The architecture was verified on the Nexys 4 FPGA board, and the processor itself can process 640 × 480 (VGA) @ 16 fps video stream, while using only 24 mW of energy.

In conclusion, there are multiple solutions working in real time on the FPGA platform, whose crucial parameters are gathered in Table 1. However, most of the implementations were realised for the single-scale version of the LK algorithm, which is the limiting factor for detecting bigger pixel displacements, e.g., for fast-moving objects. Another drawback of the existing solutions is a mostly low video resolution that can be processed, reducing or even completely missing some visual information (such as small objects). The only solutions working with the most popular high-resolution cameras are [25] (Full HD) and [27] (HD), but the never-ending technological evolution hints towards even higher ones, such as 4K. Among the multi-scale versions of the LK algorithm, some existing solutions were realised as very efficient custom processors while others on FPGAs, but all working in very low resolution (VGA), often insufficient for modern applications. The closest competitor is our previous work [27], which is capable of processing two scales in HD resolution. Therefore, we propose an architecture able to process the LK algorithm in 4K resolution in real time, using multiple scales and with very small energy consumption, making it the first implementation with such parameters.

### 3.2. Horn-Schunck FPGA Implementations

One of the first works in which a hardware implementation of the Horn–Schunck algorithm on an FPGA platform was realised is [30], where the processing of 256 × 256 @ 60 fps video stream was achieved. To compute the derivatives, a “cube” of neighbouring 2 × 2 × 2 pixels was used. The third dimension of that “cube” was time, i.e., the previous and the current frame. It was verified that 50–500 iterations are usually sufficient in the HS algorithm. However, in this work, subsequent iterations were performed using consecutive pairs of frames. This approach worked well, as the differences between frames were small.

In another paper [31], a hardware implementation of the Horn–Schunck algorithm enabled the processing of 320 × 240 video stream in real time with only eight iterations. The distinguishing feature of this work was a different way of storing data in memory—the pixels for computing the derivatives were placed in different blocks of memory. This approach allowed multiple values needed for further calculations to be read out simultaneously.

In the next work [32], the processing of 256 × 256 @ 257 fps video stream was achieved. Such a high frequency was obtained by using approximate division results, which can be implemented as bit shifts, LUT tables or cascades of comparators. Additionally, different clocks were used in operation to reduce the influence of the critical path. In the solution, data buffering was also used—the pixels were transferred from an external RAM via VDMA (video direct memory access) to a FIFO (first in, first out) queue and then to the input of the HS module.

Another work is [33], where the processing of 640 × 512 @ 30 fps video stream was carried out with 30 iterations of the HS algorithm. According to the authors, the presented architecture is also capable of handling 20 fps of a 4K video stream. Small and limited flow values were assumed, which allowed the use of a LUT table to obtain division results, saving hardware resources. Two implementation approaches were also tested—iterative with the use of an external RAM and pipelined—the latter was found to provide much higher data throughput despite the higher resource utilisation.

In the work [34] an FPGA platform was used as an accelerator for cloud motion analysis using the Horn–Schunck algorithm. According to the authors, the proposed application is theoretically capable of processing video streams of up to 3750 × 3750 pixels in real time. In order to speed up the computation, the derivatives were calculated only once, and subsequent iterations of the algorithm were realised in a pipelined way to allow more data to be processed simultaneously. As in many other works, a divider was used in the form of a LUT table.

One of the works with the highest processing performance is [35], which is able to process 1920 × 1080 (Full HD) @ 60 fps video stream with 32 iterations of the single-scale HS algorithm. In this work, a pipelined architecture was also used instead of an iterative one with the use of RAM, thus significantly increasing throughput. In order to operate in real time and reduce resource utilisation, the divide operation was done only once for a pair of frames. As a part of the experiments, different masks for computing the derivatives and averaging were compared—it was found that the values of the derivatives have a key influence on the results of the entire algorithm. The authors also presented an optimised version of their solution, able to process 128 iterations at 84 fps—such a speed-up was possible after omitting the pre- and post-processing. Compared to the reference software model on a CPU, 128 iterations of the HS algorithm were executed 106× faster on the hardware platform. This result confirms the validity of using an FPGA platform for the optical flow determination in real time.

The only work in which the processing of 3840 × 2160 (4K) @ 48 fps video stream was realised is [36]. The authors used a red-black SOR solver to decrease the number of iterations for the 4-point neighbourhood. A design space exploration was performed as a part of the work, taking into account different video resolutions, the number of iterations and the values of the parameter α. The authors stated that α value of 5 is usually a good choice, while 10–15 iterations of the HS algorithm are sufficient, taking into account the increasing resource utilisation with each iteration. Different versions of the implementation were prepared—high precision or high throughput. The algorithm runs on the Xilinx VC707 platform using 11.27 W.

In terms of the multi-scale HS algorithm, the number of publications is significantly smaller. One of them is [37], in which the authors achieved the processing of 1920 × 1080 (Full HD) @ 60 fps video stream, using 4 custom WXGA processors (1280 × 800 pixels). The architecture of a 2-scale algorithm using SOR solver was verified on the Xilinx Kintex-7 FPGA platform. Among the improvements proposed by the authors are the modified multi-scale method, initial value creation, and adaptive acceleration parameter selection. In the work, different numbers of iterations were used for the scales—32 for the smaller one and 8 for the bigger one. The authors also stated that their solution can possibly process a 4K video stream, but 16 processors would be needed.

Another implementation of the multi-scale HS algorithm, able to process 1024 × 1024 @ 29 fps video stream, was presented in [38]. The authors used 3 scales and, similarly to the previous work, different numbers of iterations for different pyramid levels—precisely 20, 10 and 5 for the biggest one. As a part of the work, a design space exploration was done, taking into account different precisions (16 and 32 bits), interpolation methods (bilinear and bicubic) and processing schemes (iterative, partial pipeline, fully pipeline, and fully pipeline parallel). The last processing scheme allows for calculating multiple neighbouring pixels at the same time by duplicating necessary modules. However, it is not clear whether any optimisations were made to avoid data redundancy.

In summary, just like in the case of the LK algorithm, there are multiple solutions of the HS algorithm that work in real time on FPGA platforms, the most important parameters of which are gathered in Table 2. Works such as [33,34,36] were realised for ultra high-resolution video streams, enabling the detection of smaller objects, but they lack in the detection of bigger pixel displacements because of the used single-scale approach. Moreover, in these implementations, the number of iterations of the HS algorithm is low due to the high use of resources in every iteration. Among multi-scale versions of the HS algorithm, only two works exist to our knowledge—one realised on an efficient custom processor working in Full HD resolution and the other in uncommon, similar to HD resolution, but with redundant operations on neighbouring pixels. Thus, we propose an architecture able to process the HS algorithm in 4K resolution in real-time using multiple scales, with high efficiency and very small energy consumption. We also do not use a red-black SOR solver like in [36], as in our implementation, we update the flow values based on the 8-point neighbourhood. Compared with existing 4K solutions, we use a similar number of iterations, but in multiple scales, along with additional modules responsible for scaling, warping and data synchronisation between the scales, making it the first implementation with such parameters.

## 4. The Proposed OF System

To implement and test particular components of the LK and HS algorithms, we used the Xilinx Zynq UltraScale+ MPSoC device, available on a ZCU 104 platform. To generate a video pass-through, we followed the steps of the Xilinx example design (https://docs.xilinx.com/r/3.1-English/pg235-v-hdmi-tx-ss/Example-Design (accessed on 30 May 2022)). The input video signal was transmitted from the computer via the HDMI 2.0 (High Definition Multimedia Interface) interface. Computations in the video pass-through were performed mainly using FPGA, but the ARM processor was also used, e.g., to control the transfer of a previous frame with the external RAM memory using the AXI4-Stream interface. The output image was sent via HDMI interface to a 4K monitor to visualise the results. Figure 3 shows a simplified scheme of the system architecture used to run the algorithms on the target platform. The 4ppc data format (4 pixels per clock) was used, reducing the minimal frequency required for real-time processing in 4K resolution to 150 MHz—more details about the data format are presented in the subsection below. However, due to buffering during the data exchange with the external RAM, we had to increase this value to 300 MHz. All of the modules described in this section were developed using SystemVerilog HDL in Vivado Design Suite IDE 2020.2.

### 4.1. Video Processing in 4K

As already pointed out in Section 1, the real-time processing of a 4K video stream is a challenging task, especially in the case of complex or multiple calculations on each pixel, just like in gradient-based optical flow algorithms. Due to the huge amount of data that needs to be processed every second, it is generally impossible to run them on a typical CPU in real time. Therefore, GPUs, FPGAs or even dedicated hardware solutions are used, which can process data in a parallel manner. FPGAs are much more energy efficient, more flexible in programming and do not require a host computer, like GPUs. They are also far more easy to use than a custom hardware (e.g., VLSI). However, the processing of a 4K video stream @ 60 fps requires a pixel clock frequency of nearly 600 MHz, which is close to the limit of current FPGA platforms. Using such a high frequency may also cause undesirable routing and timing problems.

Therefore, the processing of multiple pixels per clock cycle can be considered—if two pixels appear on the input at the same time, two pixels must be output simultaneously. In this case we can lower the clock frequency to around 300 MHz. If we put four pixels on the input, we need to process and output four values at the same time, reducing the clock frequency to 150 MHz and so on. The data format, regarding the number of pixels processed at the same time, is denoted as Xppc (X pixels per clock) and is also referred to as vector format.

In case of a pixel-wise processing, X times more resources are needed to compute X pixel values, e.g., during conversion from RGB to greyscale. However, in the case of a context operation, it is more difficult, as neighbouring pixels may appear in the same clock cycle, in the previous one or in the next one. Therefore, a joint context has to be created to later select X contexts for X pixels, just like in the work of [39]. A typical context generation of size N×N px requires $N^{2}$ registers and N−1 delay lines, just like that presented in Figure 4a. For the Xppc format, all registers and delay lines have X times more values, while the joint context is of size (N+X−1)×N px as in Figure 4b. As a consequence of the data format used, some additional logic is necessary to perform context operations, reusing pixels and intermediate values which belong to multiple contexts to avoid redundant computations. The effective implementation of context operations has a big effect on the performance and resource utilisation, as they are essential components of both the LK and HS algorithms.

### 4.2. Optical Flow Algorithms

Apart from input/output video signals and communication with external RAM, some components were used in both algorithms. The first step of the processing was the conversion of the input frame from RGB colour space to greyscale. Next, a Gaussian blur with a commonly used mask [1,4,6,4,1]/16×[1,4,6,4,1]/16 was applied to smooth the image. Then, two frames were needed simultaneously to compute the spatial and temporal derivatives. The current frame, coming from the video input, was fed to the derivatives module and written to external RAM, while the previous frame was read from RAM and sent to the derivatives module. Another solution was also tested, which involved writing the raw image (before Gaussian smoothing) to memory. However, this approach required performing the blur on two frames at the same time, increasing resource usage, so it was discarded. Derivatives calculation and further steps were performed differently for the LK and HS algorithms, and they are described in respective subsections.

In both algorithms, the resulting flow was expressed using 13 bits—1 for the sign, 4 for the integer part, and 8 for the fractional part. This data format was justified by the assumption of a small pixel displacement, so the maximum flow was limited to the range (−16,+16) pixels. If the flow during the calculations (in the LK or HS) exceeded ±16, it was set to a maximum value (with a proper sign). After testing different numbers of bits for the fractional part, we concluded that 8 bits ensures a good compromise between accuracy and the resulting hardware utilisation.

After obtaining the flow, the median filtering was applied in the window of 5 × 5 px to remove individual outliers and improve the results. For this task, the Batcher odd–even mergesort [40] was used separately for both flow components (each for 25 elements). Its operation is based on comparing neighbouring values and merging smaller sorted subsets. In the case of a hardware implementation on the FPGA platform, a parallelisation of comparisons between multiple values is possible, which is undoubtedly a big advantage of this approach. After 17 clock cycles, the middle element of a 25-element array was selected and output from the median module.

Another component, which is common for both implemented algorithms, is a visualisation, i.e., graphical representation of the displacement obtained for each pixel. The most intuitive approach is to draw vectors on the original image whose directions and lengths determine the directions and magnitudes of the pixel movement. In the case of a hardware implementation, visualising a dense flow in this way can be problematic and difficult to comprehend. Therefore, another method, based on HSV colour space representation, was used, in which the colour corresponds to the direction of the pixel’s movement, while the saturation is related to its speed in pixels per frame. To interpret the results presented in this way, a so-called “colour wheel” as in Figure 5a is used.

The first step of the visualisation was to calculate the angle of the vector formed by the optical flow components *u* and *v* with the horizontal axis for each pixel. This operation requires arctangent calculation, which is costly in terms of hardware resources; therefore, a LUT table was prepared with pre-calculated values for all possible input vectors. Taking into account the accuracy of the computations and the utilisation of hardware resources, it was decided to set the number of values stored in the array to 2048, which translates into an 11-bit input vector (the ratio between vertical and horizontal flow). The output of the LUT table was set to a 10-bit angle in range $\left[0^{\circ },90^{\circ }\right)$. Taking into account signs of both flow components, the resulting angle (H component) was in the range $\left[0^{\circ },360^{\circ }\right)$. The use of LUT table for the limited range allowed to reduce the use of hardware resources and avoid redundant data storage.

In addition to the vector angle, it was also necessary to determine its length using the Euclidean norm. In this work, the vector lengths were limited to fit in the [0,1] range of the S component of the HSV colour space. However, if the maximum possible displacement is known, a normalisation can be used. The third component (V) was set to the maximum value for all pixels. Finally, to display the results on the monitor, a conversion to RGB colour space was needed. A visualised optical flow (precisely the ground truth from the MIT CSAIL *Camera motion* sequence [1]) is shown in Figure 5b. For easier perception on the monitor, the S and V components can be swapped, which results in black colour for static background instead of white, without any effect on the visualised direction or speed of the pixels.

#### 4.2.1. Implementation of the Lucas-Kanade Algorithm

The first step that distinguishes between the LK and HS algorithms is the way of computing the spatial derivatives *I_x_*, *I_y_* on a previous frame (from RAM) and the temporal derivative *I_t_* between the previous frame and the current one, coming from the camera. Various masks for the spatial derivative in both directions were analysed, including [−1,0,1]/2 and [−1,8,0,−8,1]/12. After testing them on a few sequences, it turned out that the first one ensures slightly better accuracy. In the case of a hardware implementation, it avoids the division operation, limiting the resource utilisation and eliminating rounding errors, and also reduces the latency of the module, which is even more important in the case of a multi-scale version. In the case of the temporal derivative, the simplest subtraction between the frames was realised. For all three derivatives, additional thresholding was used—if the result was small (e.g., below 5), it was zeroed. This was motivated by the noise that occurs in the source video signal. Finally, the derivatives calculated for each pixel were output simultaneously. A simplified scheme for these calculations (for clarity for one processed pixel, which translates to the 1ppc mode) is presented in Figure 6. In general case, X contexts are generated for X pixels during derivative calculations, as in Figure 4b.

In the next step of the LK algorithm, the derivatives are multiplied, generating 5 values for each pixel as in Equation (Equation 14), which are later used for context generation and summation in the window of size N×N px. Choosing the right *N* is a difficult task—on one hand, a smaller window size allows to determine optical flow with more details (such as small, thin objects) and to reduce resource utilisation, but with more errors. On the other hand, a larger window size results in smoother flow (especially for bigger objects or high video resolution), but at the cost of increased resource utilisation, latency, and sometimes undesirable smoothness. Therefore, as a compromise, *N* was set to 9 because of the used multi-scale approach, which allows bigger displacements detection.
(14)$$\begin{array}{cc} \; & I_{xx}=I_{x}\cdot I_{x}\\ \; & I_{xy}=I_{x}\cdot I_{y}\\ \; & I_{yy}=I_{y}\cdot I_{y}\\ \; & I_{tx}=I_{t}\cdot I_{x}\\ \; & I_{ty}=I_{t}\cdot I_{y} \end{array}$$

Due to the data format used (4ppc), many identical summations were performed multiple times (for different “central” pixels), which increased the resource usage. Precisely, after derivatives multiplication, a joint context of size 9 × 12 px was generated; however, a 9 × 6 px window was common for all 4 pixels. Therefore, they could be summed only once and later added to the rest of the context (remaining 9 × 3 px windows for different pixels). Due to the avoidance of redundancy, the number of summations was limited by 50%, significantly reducing hardware utilisation. Moreover, some of the summators were implemented with LUTs and flip-flops, while others with DSPs to balance the usage of different resource types, avoiding congestions and routing problems.

A simplified schematic of this method (for one product, e.g., *I_xx_*) is presented in Figure 7. Weights *w_i_* from Equation (Equation 12) were all assigned 1 to smooth the flow and reduce the influence of erroneous “central” pixels. Another solution was also implemented and tested—generating context for incoming derivatives and then performing their multiplication. This method significantly reduced BRAM usage, as the number of bits per pixel was much smaller (27 vs. 85), but at a cost of considerably higher DSP/LUT/FF utilisation. This approach, even with an effective split of the used resource types, resulted in congestions and routing problems during the implementation of the multi-scale version, and thus it was not used in the presented solution.

To obtain the optical flow for a given pixel, it is necessary to calculate Equation (Equation 13). Equation (Equation 12) can be rewritten in the form of Equation (Equation 15), using $A_{11}=\sum I_{xx}$, $A_{12}=A_{21}=\sum I_{xy}$, $A_{22}=\sum I_{yy}$, $b_{1}=\sum I_{tx}$ and $b_{2}=\sum I_{ty}$. To solve this equation, the summation matrix has to be inverted, which requires matrix determinant calculation, as in Equation (Equation 16). Finally, the flow can be expressed as in Equation (Equation 17).

This calculation is very expensive in terms of resource utilisation on the FPGA platform, as it requires a division operation. We used a fixed-point architecture; therefore, dividends and divisors are big (46 bits each). However, to save resources, we only performed the division operation once for each pixel, calculating the inverse of detA and then using the result in multiplications. In this way, our dividend data width is 2, the divisor width is 46 and the result is expressed using 64 bits. This solution slightly degrades accuracy more than in the case of division of the 46-bit dividend and 46-bit divisor, but the resource usage is reduced significantly.
(15)$$\left(\begin{array}{cc} A_{11} & A_{12}\\ A_{21} & A_{22} \end{array}\right)\left(\begin{array}{c} u\\ v \end{array}\right)=-\left(\begin{array}{c} b_{1}\\ b_{2} \end{array}\right)$$
(16)$$\det A=A_{11}\cdot A_{22}-A_{12}\cdot A_{21}$$
(17)$$\left(\begin{array}{c} u\\ v \end{array}\right)=-\left(\begin{array}{cc} A_{22} & -A_{12}\\ -A_{21} & A_{11} \end{array}\right)\cdot \left(\begin{array}{c} b_{1}\\ b_{2} \end{array}\right)\cdot \frac{1}{\det A}$$

As noted in Section 2.2, sometimes additional thresholding based on the eigenvalues of the summation matrix is used to remove possibly incorrect results. However, this operation requires several multiplications and calculation of a square root for each pixel, which considerably increases resource utilisation. It is also difficult to specify the appropriate value of the threshold, especially when using the multi-scale method. Therefore, we decided not to use any additional thresholding of the results. Figure 8 shows a simplified block diagram of the implemented Lucas–Kanade optical flow determination algorithm in the single-scale version.

#### 4.2.2. Implementation of the Horn–Schunck Algorithm

A different set of operations is needed to obtain the optical flow according to the HS algorithm. The first step requires the calculation of derivatives—spatial *I_x_*, *I_y_* and temporal *I_t_*. For this task, a context of size 2 × 2 × 2 px has to be generated with the time as the third dimension (previous and current frame). Therefore, for both images a context of size 2 × 2 px is generated. Marking the previous frame (from RAM) as *I*_1_ and the current one as *I*_2_, the derivatives are calculated according to Equations (Equation 18)–(Equation 20). Similar as in the LK method, an additional thresholding was used to zero the values which were smaller than a set parameter. A scheme of derivatives calculation in the HS algorithm is shown in Figure 9.
(18)$$\begin{array}{cc} I_{x}=\frac{1}{4} & [I_{1}\left(i,j+1\right)+I_{1}\left(i+1,j+1\right)+I_{2}\left(i,j+1\right)+I_{2}\left(i+1,j+1\right)\\ \; & -I_{1}\left(i,j\right)-I_{1}\left(i+1,j\right)-I_{2}\left(i,j\right)-I_{2}\left(i+1,j\right)] \end{array}$$
(19)$$\begin{array}{cc} I_{y}=\frac{1}{4} & [I_{1}\left(i+1,j\right)+I_{1}\left(i+1,j+1\right)+I_{2}\left(i+1,j\right)+I_{2}\left(i+1,j+1\right)\\ \; & -I_{1}\left(i,j\right)-I_{1}\left(i,j+1\right)-I_{2}\left(i,j\right)-I_{2}\left(i,j+1\right)] \end{array}$$
(20)$$\begin{array}{cc} I_{t}=\frac{1}{4} & [I_{2}\left(i,j\right)+I_{2}\left(i,j+1\right)+I_{2}\left(i+1,j\right)+I_{2}\left(i+1,j+1\right)\\ \; & -I_{1}\left(i,j\right)-I_{1}\left(i,j+1\right)-I_{1}\left(i+1,j\right)-I_{1}\left(i+1,j+1\right)] \end{array}$$

Calculation of the optical flow is done differently in the HS method—first, the flow is initialised and then it is updated in an iterative manner based on neighbouring pixels. Therefore, the number of iterations is one of the crucial parameters that affects the accuracy of the results. However, the FPGA platform works very well in the case of pipelined data processing instead of an iterative one. To perform multiple HS iterations in a pipelined way, several identical modules for the flow calculation have to be generated and connected with each other, which is shown in Figure 10a. The other solution, not chosen in this work, which requires re-using the same module and the external RAM to store intermediate flow values, is shown in Figure 10b.

In the flow initialisation step, Equations (Equation 6) and (Equation 7) were used. The denominator in these equations is common and used in further iterative flow refinement, so to avoid redundant operations, this value is calculated only once in the initialisation step and marked as ψ as in Equation (Equation 21). 20 bits are allocated for it—1 for the integer part and 19 for the fractional part, as the possible values belong to (0,1] range for integer α values. In subsequent calculations, ψ value can be used in less resource-intensive multiplications than division operations in every iteration.

Choosing the right value of α parameter, responsible for the flow regularisation, is not an easy task—large α generates smooth results but blurs objects, while small α increases the susceptibility to the occurrence of incorrect values. After testing several possibilities, it was set to 3, but depending on the expected outcome (i.e., smoother flow) or the sequence used, the value of this parameter can be easily modified to obtain better results. The value of ψ was then used to compute the initial optical flow *u*_0_ and *v*_0_ from Equations (Equation 6) and (Equation 7) with average velocities set to 0, which resulted in Equations (Equation 22) and (Equation 23). The obtained values of ψ, *u*_0_, *v*_0_ and delayed derivatives *I_x_*, *I_y_* and *I_t_* were output simultaneously from the initialisation module.
(21)$$\psi =\frac{1}{\alpha^{2}+I_{x}^{2}+I_{y}^{2}}$$
(22)$$u_{0}=-\psi I_{x}I_{t}$$
(23)$$v_{0}=-\psi I_{y}I_{t}$$

In the next step, these values were used in the flow refinement procedure, using multiple identical pipelined modules. In every iteration, a context of size 3 × 3 px was generated for both flow components. Two convolutional masks for averaging were used and compared—[0,1,0;1,0,1;0,1,0]/4 and [1,2,1;2,0,2;1,2,1]/12. The latter provided slightly faster convergence, even though it used a bit more resources. To limit their usage, division by 12 was replaced by multiplication by its inverse stored in 10 bits. The average flow in a small neighbourhood (for both components) was then used for calculating the updated flow according to the HS equations, which, after using ψ, take the form of Equations (Equation 24) and (Equation 25). The expression in parentheses was common for both components, so it was calculated only once in each iteration.
(24)$$u_{n+1}=\bar{u_{n}}-\psi I_{x}\left(I_{x}\bar{u_{n}}+I_{y}\bar{v_{n}}+I_{t}\right)$$
(25)$$v_{n+1}=\bar{v_{n}}-\psi I_{y}\left(I_{x}\bar{u_{n}}+I_{y}\bar{v_{n}}+I_{t}\right)$$

Apart from the updated flow values, delayed input derivatives *I_x_*, *I_y_*, *I_t_* and calculated ψ were passed to the output of the module, which enabled further flow updates in subsequent iterations. In this way, the iterative flow refinement in the HS method was realised, and the number of iterations was controlled by a parameter which was initially set at 10. As already shown in the literature (e.g., [35]), the more iterations, the higher the accuracy of the results and the resource consumption. Therefore, choosing the proper number of iterations is always a compromise between accuracy and resource utilisation. The values obtained in the output of the last refinement module constituted the final result of the HS algorithm.

In the work of [35], additional thresholding of the results was applied, based on the approximate variance of the flow in 3 × 3 px window to ensure a smooth flow throughout the image. However, in this work (just like in the case of the LK algorithm) we did not use any thresholding, as it requires many additional resources. A simplified block diagram of our implementation of the Horn–Schunck optical flow determination algorithm in the single-scale version is shown in Figure 11.

### 4.3. Implementation of the Multi-Scale Method

To enable the calculation of the optical flow in many scales, several additional modules were necessary. The first of them was responsible for downscaling the image. The most common and the simplest scaling factor of 0.5 was used, i.e., each of the image dimensions was reduced twice. Precisely, downscaling was performed for two images—the current one and the previous one (from RAM). The simplest approach is to select every second pixel in every second line. However, when downscaling the image multiple times, many details are lost. Therefore, we used a bilinear interpolation—using one window of 4 pixels (2 × 2) without overlaps with neighbouring ones, we obtained one pixel value as their average.

As the number of pixels in subsequent scales decreases, we decided to use the changing data format to send and process fewer pixels at the same time. We initially used 4ppc (in scale 0, i.e., input one), but in scale 1, we changed it to 2ppc every second line, in scale 2, 1ppc every fourth line, in scale 3, 0.5ppc (one pixel per two clock cycles) every eighth line, and so on. To mark the validity of the pixels without affecting the AXI4-Stream signals, we added a *tvalid_mod* signal, which was used in the calculations instead of the original *tvalid*, which was used only for the correct visualisation of the data on the monitor. A simple schematic of the implemented downscaling method is shown in Figure 12.

To obtain the optical flow in a smaller scale (i.e., higher level of the image pyramid), one of the optical flow algorithms (LK or HS) was used. With changing data format, these modules were modified to enable OF calculation in the Xppc mode. They were both tested in a single-scale version in 4ppc, 2ppc, 1ppc and 0.5ppc formats with decreasing number of hardware resources used—detailed information is provided in Table 3 for the LK algorithm and Table 4 for the HS method. As can be seen in the tables, the utilisation of all resource types decreases with the diminishing number of pixels processed per clock cycle. However, the difference between 1ppc and 0.5ppc formats appears only in the case of Block RAM, as fewer pixels have to be stored in contexts’ delay lines—other resources are used in the same amount, but with different frequency (1 time per 1 or 2 clock cycles). In this way, *L* scales can be processed in parallel, but without using *L* times more resources.

After calculating the flow in a smaller scale, it has to be used to compensate for the motion and transform one of the images to enable processing in a larger scale. We decided to warp (transform) the current frame towards the previous one, stored in RAM. A typical solution is to upscale both flow components and perform warping on a bigger image, for which usually a bilinear interpolation is used. In terms of FPGA, two solutions exist for this operation—reading calculated pixels from RAM or generating a context. As we limited the flow to ±16, we decided to reduce the data transfers to external RAM and use the latter solution with a window of size 33 × 33 px. However, after upscaling the flow, we would need a context of 65 × 65 px in a bigger scale, which requires a lot of resources.

Therefore, we decided to perform the warping in a smaller scale and then upscale the warped image, as the difference in results accuracy between these approaches was negligible. The method chosen had significant advantages in terms of resource saving, such as smaller context (fewer lines to save in registers) and lower number of pixels processed simultaneously (e.g., generating contexts for 2 “central” pixels instead of 4, for 1 “central” pixel instead of 2, etc.). The main disadvantage was using the smaller image instead of the bigger one, thus losing some details, such as sharp object edges, but it was an acceptable cost of limiting hardware utilisation. To synchronise the image with the flow, the former was delayed using a FIFO queue. Our approach also enabled to delay the image in the 2ppc format instead of 4ppc, thus reducing twice the memory needed. The image warping was performed by bilinear interpolation—the flow helped to determine target pixel position (usually with the fractional part), and four neighbouring pixels were used for interpolation of the pixel brightness. This operation is presented schematically in Figure 13.

As mentioned above, after warping, the image has to be upscaled with the inverse factor as for downscaling, which in our implementation is 2. Again, bilinear interpolation was performed, but this time a context of size 5 × 5 px was generated with a new line of pixels showing every two lines of context. In this way, out of five context lines, two or three were valid, while the rest was invalid. This allowed us to generate the context according to the input *tvalid_mod* signal and at the same time generate one new line to insert between two input valid lines. This approach was not the most effective one, as for the interpolation, the context of 3 × 3 px could be sufficient (with two valid input lines), but it was a direct consequence of the implemented multi-scale method and changing data formats. During upscaling, the data format changes in the opposite way as in downscaling, i.e., from 2ppc format to 4ppc, from 1ppc to 2ppc, from 0.5ppc to 1ppc and so on. The validity of pixels also changes in this module, as between two valid lines, a new one is generated; therefore the *tvalid_mod* signal is modified accordingly. The scheme of the upscaling procedure is presented graphically in Figure 14—for simplicity, a context of size 3 × 3 px with two valid lines is shown instead of the full 5 × 5 px context, implemented in the hardware.

The last modules in the multi-scale method were additional FIFO queues. The first of them was used to synchronise the current frame after warping and upscaling with a delayed previous frame, which was stored in FIFO. This allowed us to compute the optical flow in a bigger scale with either the HS or LK algorithm. As mentioned before, the image after warping and upscaling is slightly blurred, and therefore a higher threshold had to be used in the derivatives calculation stage to eliminate some differences resulting from blurring.

Finally, the flows obtained at different levels of the pyramid had to be synchronised and summed to perform a visualisation of the results of the entire multi-scale algorithm. For this task, we also used FIFO queues, in which flows from the smallest scale up to the second largest were stored. The flows were then upscaled to match the size of the bottom level of the pyramid—for both flow components, we used the same module as for the image upscaling. The flow magnitudes also had to be enlarged by a certain power of 2, based on the scale. However, to suit our visualisation module, we decided to divide the flow magnitudes from the bigger scales instead of multiplying them from the smaller scales. In this way, the summed flow was still in the range ±16 and allowed visualisation with the proper relative velocity between different scales.

Figure 15 presents a simplified block scheme of the optical flow computation (for either the LK or HS algorithm) in a two-scale version. The same scheme can be used for three, four, or more scales, using the same types of modules with different parameters.

## 5. Evaluation

The optical flow obtained for a pair of images can be compared with a reference value, and the accuracy of the algorithm can be evaluated. At the same time, the use of some typical coefficients allows comparing the proposed solution with those available in the literature. Among the most popular values calculated for the optical flow is *E_AAE_* (*average angular error*), defined by Equation (Equation 26), which is the average angular error between the normalised ground truth vector (*u_r_, v_r_, 1*) and the determined (*u, v, 1*) for all *N* pixels. The second popular indicator is *E_AEE_* (*average endpoint error*), which is expressed by Equation (Equation 27). It corresponds to the average endpoint error between the obtained flow and the ground truth in pixels, calculated using the Euclidean norm. Density is also often used to denote the ratio of the number of pixels with a determined optical flow to the total number of pixels in the image. In the case of our implementation, the density is 100% (apart from the pixels without “correct” context), since the flow values for all pixels are determined and no thresholding of the results is applied.
(26)$$E_{AAE}=\frac{1}{N}\sum_{N}\arccos\frac{1+u_{r}u+v_{r}v}{\sqrt{\left(1+u_{r}^{2}+v_{r}^{2}\right)\left(1+u^{2}+v^{2}\right)}}$$
(27)$$E_{AEE}=\frac{1}{N}\sum_{N}\sqrt{{\left(u-u_{r}\right)}^{2}+{\left(v-v_{r}\right)}^{2}}$$

Video sequences from different datasets are used for the evaluation of the optical flow algorithms. One of the most popular in recent years, especially for solutions based on neural networks, is the Sintel dataset [41]. It features factors that are more demanding for optical flow determination algorithms, such as dynamic motion, reflections, occlusions, blurs and atmospheric effects. Another popular, but quite challenging one is KITTI [42]. It contains sequences recorded in traffic from the perspective of a moving vehicle.

A simpler, but very popular dataset is Middlebury [43], which is used for a detailed evaluation below. Another one worth mentioning is MIT CSAIL [1], whose authors focused on the accurate edge determination of the objects. Sequences from this dataset were used when implementing and testing various parameters of the LK and HS algorithms. Unfortunately, it is not used very often, which significantly limits the possibility of comparing results on it with other works. Nevertheless, one of the sequences, *Camera motion*, was used, for which the results are presented in Figure 16. It visually compares the flows obtained using the Lucas–Kanade and the Horn–Schunck methods with the ground truth.

### 5.1. Middlebury Dataset

Despite the publication of newer datasets in recent years, the Middlebury [43] is still widely used for evaluation, which is also the case in this work. Its main advantages are the relative simplicity and high popularity, so many published solutions can be compared with each other. The calculated errors with respect to the ground truth are included in Table 5 and Table 6. It should be emphasised that only solutions implemented on FPGA platforms are analysed—complex or deep neural network-based methods are not included in these tables. This approach is determined by the aim to compare our solution with its direct competitors rather than with a whole range of more advanced solutions, since implementation on an FPGA and obtaining real-time operation for many of them in very high resolutions is not possible with current technology. Example results for both algorithms implemented, along with selected frames from the Middlebury dataset and their ground truth, are provided in Figure 17.

Based on the results obtained, some conclusions can be drawn. First of all, the use of the multi-scale method significantly improves the accuracy of the obtained flows. Compared to other implementations, our solution yields very good results on some sequences and weaker results on others. This is due to the parameters and simplifications adopted for the hardware implementation in 4K resolution—depending on the sequence and the occurring displacements. Of course, all the methods compared return much worse results than the complex or deep neural network-based methods, but as mentioned, only real-time solutions running on an FPGA platform were compared. In the case of the HS algorithm, the key influence on the accuracy of the results has the value of α parameter and the number of iterations, which can be easily adjusted for a specific sequence. However, for the evaluation of our solution, we used the same parameters for the whole set, according to the hardware implementation—α = 3, 10 iterations in the smaller scale and 5 in the larger scale.

### 5.2. Resource Utilisation

As presented in Section 4, the proposed algorithms were run on a ZCU 104 card with the Xilinx Zynq UltraScale+ XCZU7EV-2FFVC1156 MPSoC chip. After analysing the computational resources available on this chip and their utilisation by individual system components, we limited our algorithms to using two scales. The main reason for this choice was the need to synchronise data from multiple scales, which is performed using FIFO queues. Data processing in the third scale forces the need to store a larger number of image lines (for calculations in successive scales), which in the case of 4K resolution, means 3840 values for each line. This risks very high memory resource utilisation and routing problems and sometimes even exceeding available resources and making the implementation impossible. However, if a larger FPGA is used, the architecture presented can easily be extended to more scales.

A certain amount of resources was also used for the video pass-through, enabling the correct reception and transmission of the video signal. Table 7 shows the resource utilisation for the LK algorithm—starting with the video pass-through itself, through the single-scale version, and ending with the two-scale version (with all required FIFO queues for data synchronisation). For the multi-scale version, it was necessary to reduce the number of BRAM modules used, as they slightly exceeded the number of elements available in the chip. Only one modification was made—the context dimensions for the warping stage were reduced from 33 × 33 px to 25 × 25 px, allowing the entire algorithm to fit on the target platform. However, a hardware platform with more memory is required to enable warping in a larger context or to use more scales.

An analogous summary for the HS algorithm is provided in Table 8, with 10 iterations of the algorithm performed in the single-scale version, and 10 iterations in the smaller scale and 5 in the larger scale in the two-scale version. The dimensions of the context at the warping stage were changed in the same way as in the LK algorithm. A larger number of iterations of the HS algorithm can be achieved by using a chip with more internal memory resources. Another way to achieve this could be to partially use external memory to store intermediate results.

In the case of the LK algorithm, the window summation operation required most resources, but its size largely determines the accuracy of the results. In the case of the HS algorithm, most costly was the iterative updating of the flow, but this element has the greatest impact on the accuracy of the resulting flow. Therefore, for both algorithms, the solution presented is a compromise between the accuracy of the results and the amount of resources used. If one of these factors is more important, then it is possible (to some extent) to make simple modifications to the architecture at the expense of the other factor. However, in both algorithms, the most costly operations are those associated with synchronising data from multiple scales using FIFO queues—but in a multi-scale version of the algorithm, they cannot be avoided. This issue is especially challenging in the case of a 4K resolution, which requires a lot of data to be temporarily stored, so it can be perceived as the bottleneck of the proposed solution.

FPGA platforms operate at low power consumption while maintaining high performance in data processing. The architecture described consumes only 5.29 W for the single-scale version of the LK algorithm and 5.70 W for the multi-scale version according to the estimation in the Vivado Design Suite IDE. For the HS algorithm, it consumes 5.35 W for the single-scale version and 5.70 W for the multi-scale version, also according to Vivado’s estimation. This is about 2× less than the single-scale implementation in 4K resolution with 15 iterations (11.27 W) in the work [36]. Thus, our solution can process multiple scales with a much lower energy requirement.

In terms of performance, the multi-scale version of the LK algorithm achieves 463 GOPS (giga operations per second), which translates to 81 GOPS/W. The multi-scale version of the HS algorithm achieves 446 GOPS, which equals 78 GOPS/W. Both values are much higher than those in ref. [36], where the authors achieved 43 GOPS/W for the processing of a 4K resolution video in one scale. These results confirm the efficiency of the proposed implementation and the advantages of using SoC FPGA devices in embedded computing. The photo of the proposed hardware system for calculating optical flow is presented in Figure 18.

## 6. Conclusions

In this work, the Lucas–Kanade and the Horn–Schunck optical flow determination algorithms were implemented on a SoC FPGA platform. To enable the detection of larger pixel displacements, a multi-scale approach was used in both methods. The applied vector data format allowed us to lower the clock rate and process the video stream in 4K resolution (3840 × 2160 pixels) in real time (60 fps). To the best of our knowledge, solutions to determine the optical flow in such high resolution in multiple scales have not been described so far in the literature (05.2022). An additional advantage of the proposed approach is its energy efficiency—the selected hardware platform consumes 5–6 W, depending on the algorithm version. An efficient processing of the multi-scale method was achieved, using different data formats depending on the scale, which allowed a significant reduction in the hardware resources used, while keeping the implementation simple.

Due to the hardware platform used—ZCU 104 with the Xilinx Zynq UltraScale+ MPSoC device, the algorithms implemented are not very complex. On the other hand, this modern device allows the processing of high-resolution data, which was not possible in the case of older implementations (described in Section 3.1 and Section 3.2), due to the insufficient amount of computing resources or memory. However, compared with other modern solutions, such as works [33,35,36,38], where the authors used devices with similar amounts of resources, we applied a more efficient way of processing data by using the vector format. Furthermore, it was changing depending on the scale, thus allowing to significantly save computing resources in the multi-scale version of the proposed architecture.

As a part of further work, it is possible to use another FPGA device with more hardware resources, mainly memory. This would allow us to extend the algorithm with additional scales and detect even larger pixel displacements, which is important in processing such a high-resolution video stream. For the HS algorithm, it would also be possible to add more iterations to obtain more accurate results. A potential idea for comparison with the current implementation could also be the operation of different scales at different frequencies—on one hand, this should reduce the overall energy consumption, on the other hand, it would introduce additional complications and difficulties in synchronising data from different scales.

The optical flow determination algorithms realised in this work could be a component of a larger system, when a hardware platform with more memory or a single-scale version of the algorithm is used, or the resolution of the processed video stream is lower. Information about the direction and speed of the pixel movement can be aggregated and, for example, allow the detection of an entire moving object. Such information is very valuable in all types of autonomous vehicles, including aerial vehicles, i.e., drones. It then enables them to detect dangerous (collision) spots while moving and avoid obstacles, ensuring safe operation in the environment. Such solutions often use high-resolution cameras, but require obtaining real-time information about the movement of the objects in order to react to them quickly enough. Therefore, algorithms realised on the FPGA platform allow to meet these requirements, enabling correct and safe detection of the objects or the obstacles. Potential applications could also include video surveillance and monitoring systems to detect objects moving in the monitored space.

## Figures and Tables

**Figure 1 sensors-22-05017-f001:**
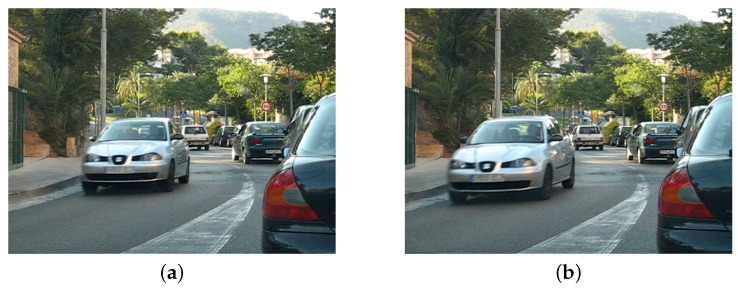
Exemplary frames from *Camera motion* sequence [1]. (**a**) Previous frame; (**b**) next frame.

**Figure 2 sensors-22-05017-f002:**
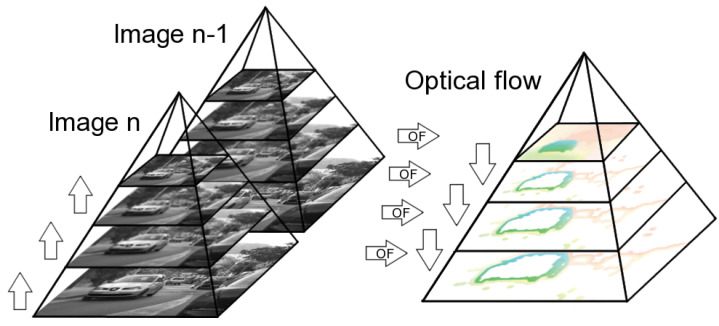
Optical flow calculation scheme in the multi-scale method. An image pyramid is generated for both frames from the sequence. Then, the optical flow is determined in the smallest scale, and its results are used to modify the previous frame to perform the motion compensation. Next, the optical flow is calculated in a bigger scale and the motion compensation is performed again—this procedure continues to the biggest scale.

**Figure 3 sensors-22-05017-f003:**
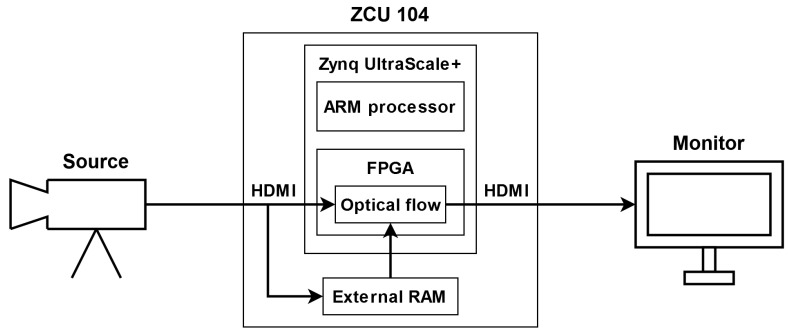
Simplified scheme of the system architecture.

**Figure 4 sensors-22-05017-f004:**
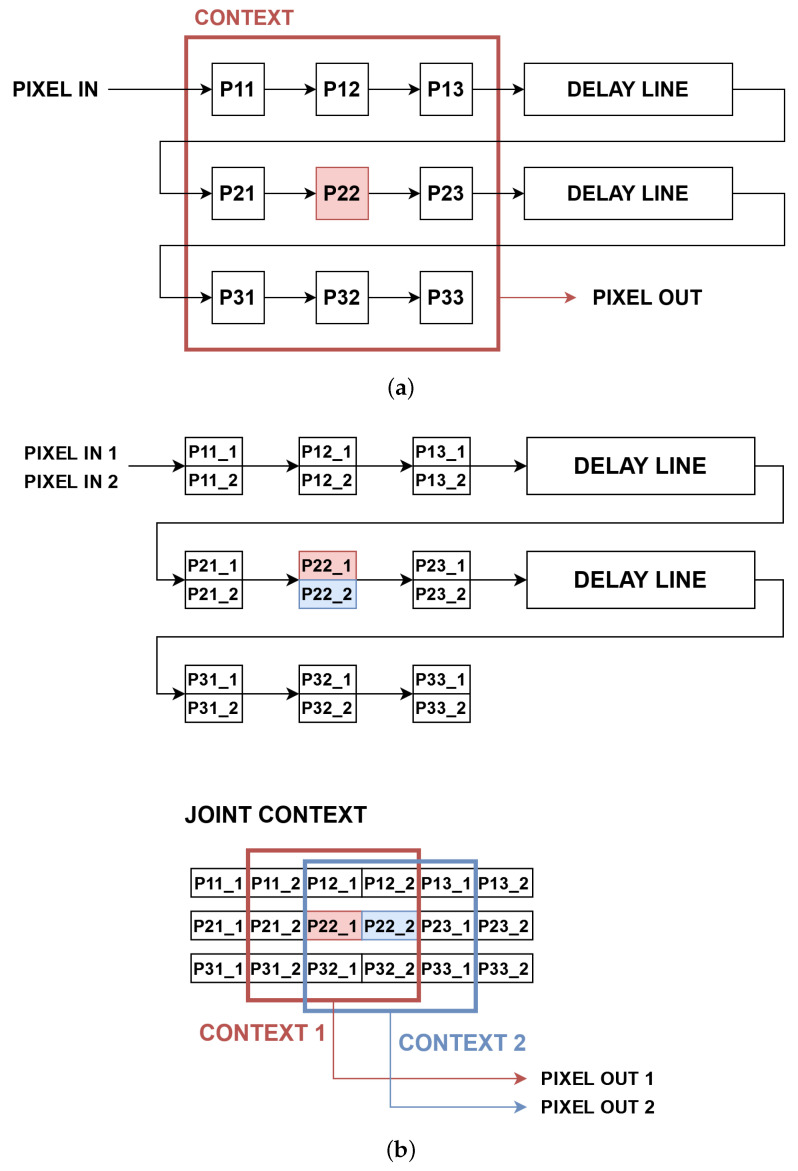
Comparison of context generation for different data formats. The 2ppc format can be generalised to any Xppc. (**a**) Context generation for the 1ppc format; (**b**) context generation for the 2ppc format. Two central pixels are coloured in red and blue.

**Figure 5 sensors-22-05017-f005:**
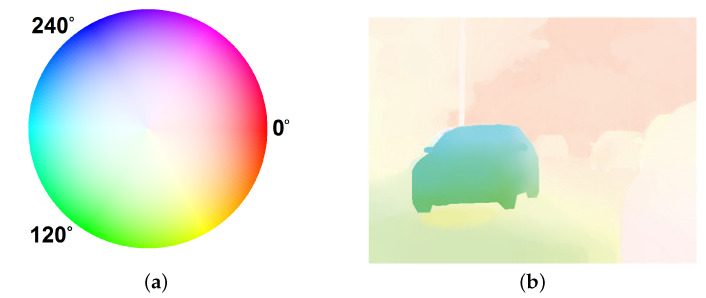
The optical flow visualisation method used. (**a**) Colour wheel; (**b**) exemplary result.

**Figure 6 sensors-22-05017-f006:**
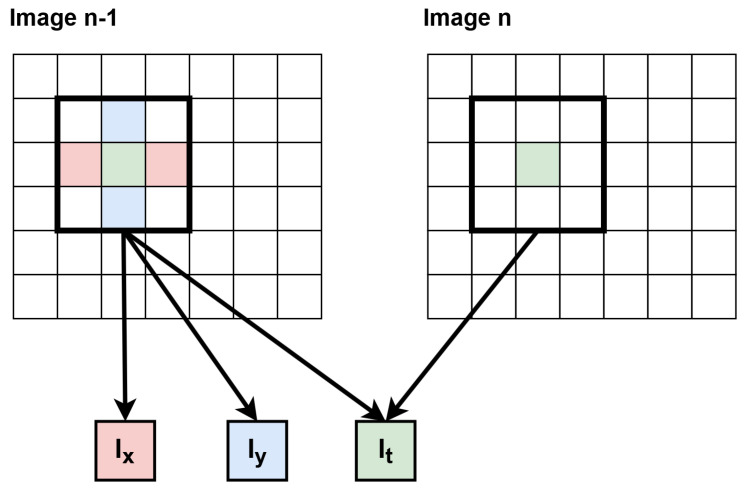
Scheme of calculating derivatives in the LK algorithm. Using context of size 3 × 3 px, 3 derivatives are calculated for one pixel. Image n−1 denotes the previous frame, while Image n denotes the current one.

**Figure 7 sensors-22-05017-f007:**
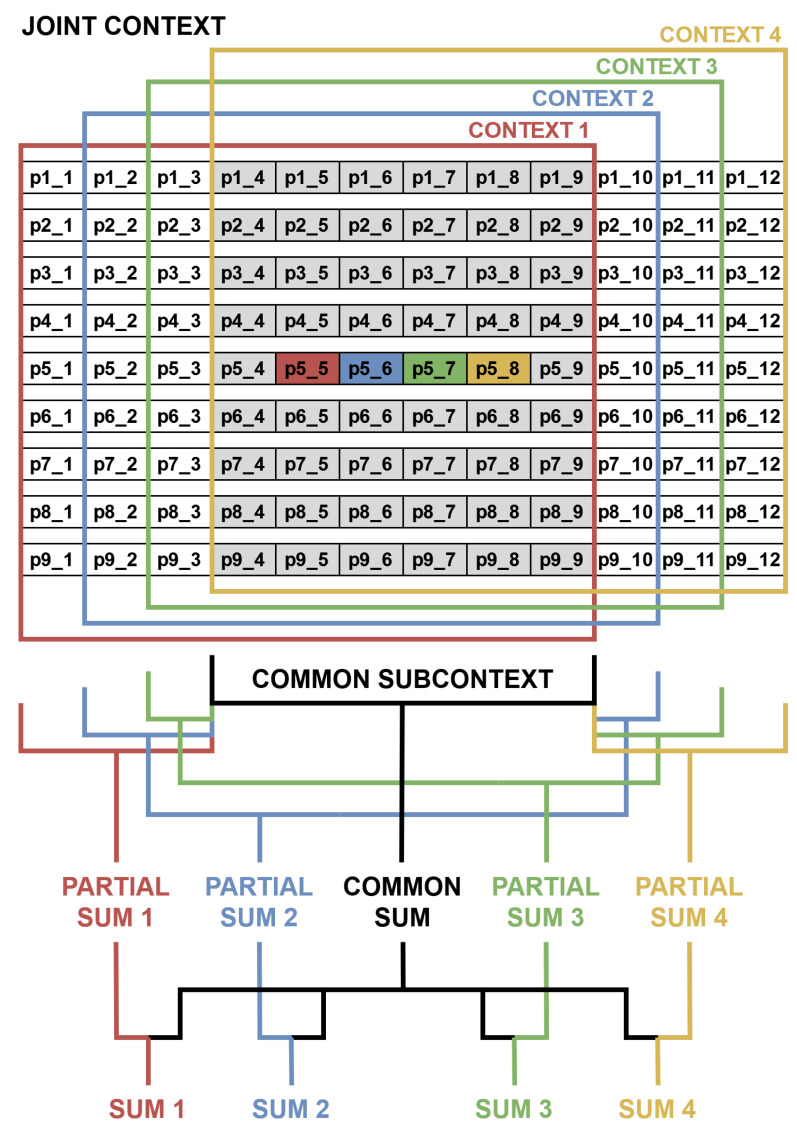
Scheme of the summation method for the 4ppc data format. The sum of non-white pixels is calculated only once for the joint context, saving hardware resources.

**Figure 8 sensors-22-05017-f008:**
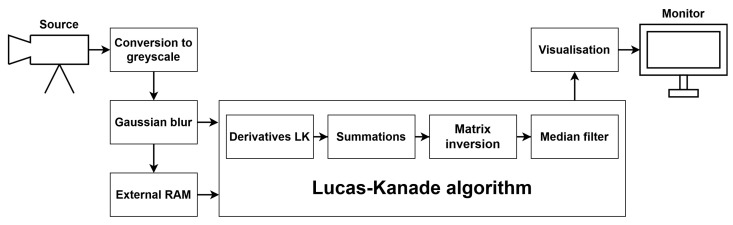
Block scheme of the single-scale LK algorithm.

**Figure 9 sensors-22-05017-f009:**
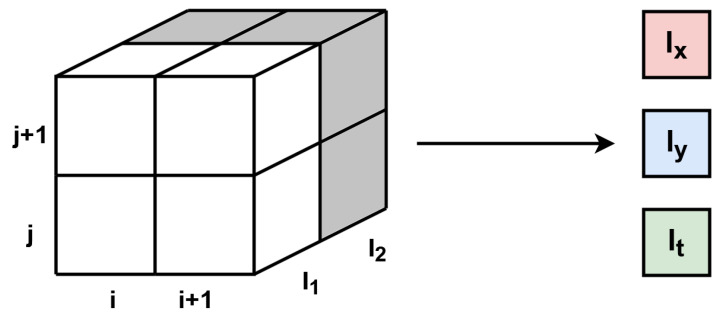
Derivatives calculation scheme in the HS algorithm. White pixels are from *I*_1_ (previous frame), while grey pixels are from *I*_2_ (current frame). Using cube of size 2 × 2 × 2 px, 3 derivatives are calculated for one pixel.

**Figure 10 sensors-22-05017-f010:**
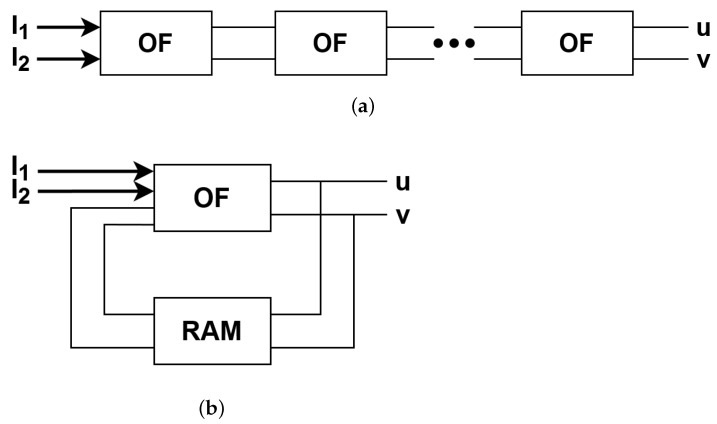
Methods for processing multiple iterations of the HS algorithm. (**a**) Pipeline approach; (**b**) iterative approach.

**Figure 11 sensors-22-05017-f011:**
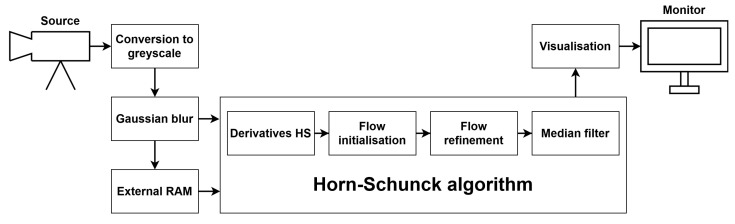
Block scheme of the single-scale HS algorithm.

**Figure 12 sensors-22-05017-f012:**
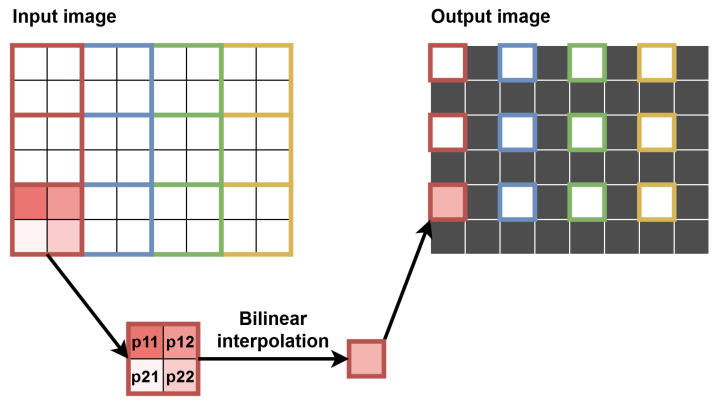
Implemented downscaling method. Black pixels in the output image are invalid, i.e., their *tvalid_mod* signal is set to 0, so they are not processed in the smaller scale. Blue, green and gold contexts are next ones being processed in the same way.

**Figure 13 sensors-22-05017-f013:**
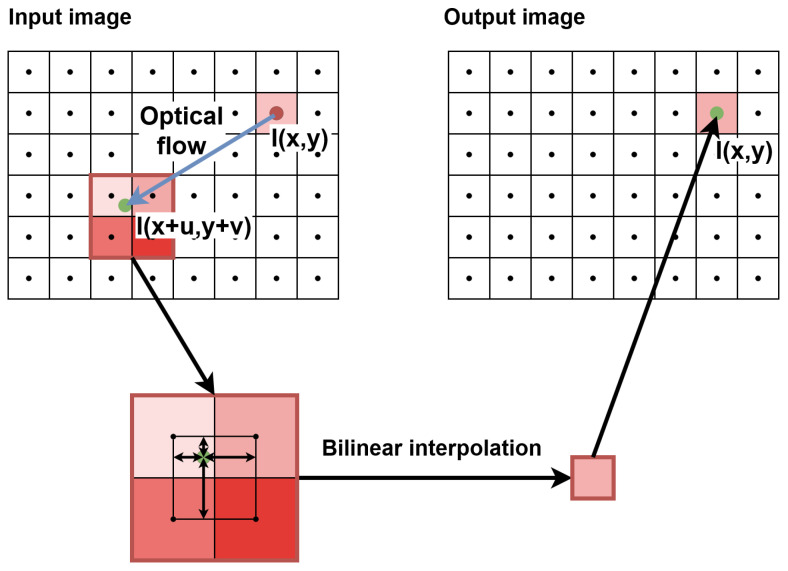
Warping of the image based on the calculated optical flow. The dark red circle in the input image is the processed pixel, the blue line shows its movement (optical flow), the green circle determines the pixel’s target position (with the fractional parts), while black dots represent “pixel centres” (for the purpose of visualising fractionals). A context of 2 × 2 px is generated for bilinear interpolation, and the resulting pixel brightness is assigned to the processed pixel in the output image.

**Figure 14 sensors-22-05017-f014:**
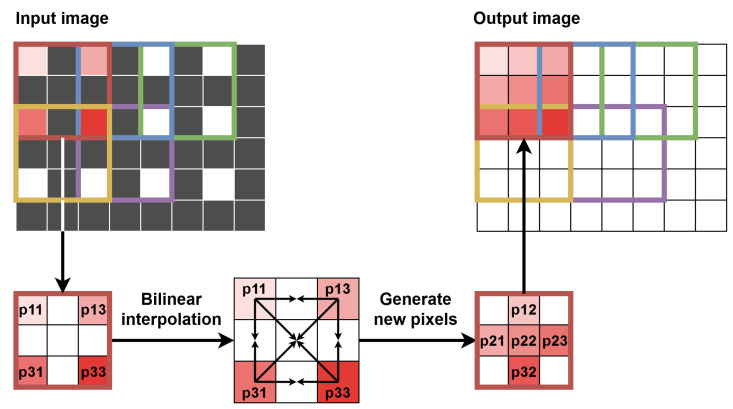
Implemented upscaling method. Black pixels in the input image are invalid, i.e., their *tvalid_mod* signal is set to 0 and they were not processed in the smaller scale. A context of 2 × 2 valid pixels is generated for the bilinear interpolation to calculate new valid pixels, which are put in relevant places of the output image. Blue, green, gold and purple contexts are next ones processed in the same way.

**Figure 15 sensors-22-05017-f015:**
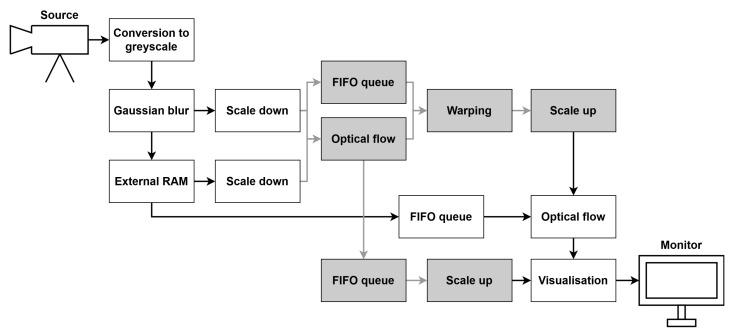
Block scheme of a multi-scale optical flow computation. In case of our implementation, white modules work in the 4ppc format and grey in the 2ppc format. The same meaning is for black and grey arrows, signifying data transfer in the 4ppc or 2ppc format. The optical flow module can be the LK method as in Section 4.2.1, the HS method as in Section 4.2.2 or in a general case, another algorithm.

**Figure 16 sensors-22-05017-f016:**
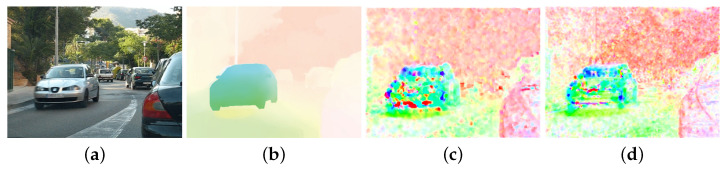
Exemplary results for the sequence *Camera motion* from the MIT CSAIL database. The images show: (**a**) a frame from the sequence, (**b**) the ground truth, (**c**) the result of the LK algorithm, (**d**) the result of the HS algorithm.

**Figure 17 sensors-22-05017-f017:**
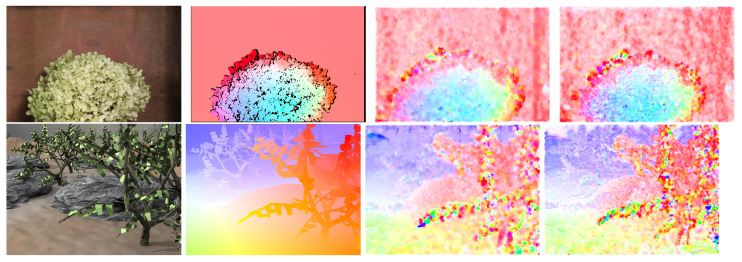
Exemplary results on Middlebury dataset sequences. The following rows contain the sequences *Hydrangea* and *Grove3*. The columns show a frame from the sequence, the ground truth, the result of the LK algorithm and the result of the HS algorithm.

**Figure 18 sensors-22-05017-f018:**
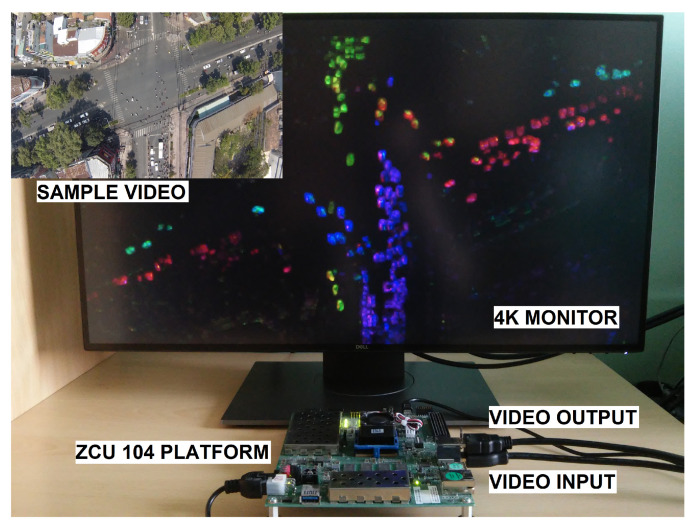
The photo of the proposed optical flow system. The input video signal is transmitted from the computer (the source) and processed in real-time on a ZCU 104 platform equipped with the Xilinx Zynq UltraScale+ MPSoC device. The calculated optical flow is transmitted and visualised on a 4K resolution monitor. The video being processed shows traffic at the intersection seen from above (top left corner of the image). Different colours relate to various directions of the moving objects.

**Table 1 sensors-22-05017-t001:** Hardware implementations of the LK algorithm on FPGA platform. Our solution works for the highest video resolution compared to these available in the literature. It also uses the multi-scale approach, which was realised only for VGA and HD resolutions.

Implementation	Scales	Resolution	FPS	Platform
Diaz [20]	1	320 × 240	30	Xilinx Virtex 2000-E
Diaz [21]	1	800 × 600	170	Xilinx Virtex II XC2V6000-4
Barranco [22]	1	640 × 480	270	Xilinx Virtex4 XC4vfx100
Kalyan [23]	1	1200 × 680	500	Altera Cyclone II
Seong [24]	1	800 × 600	196	Xilinx Virtex-6 LX760
Bagni [25]	1	1920 × 1080	123	Xilinx Zynq 7045-2
Murachi [28]	3	640 × 480	30	Custom 90nm CMOS
Smets [29]	3	640 × 480	16	Custom 40nm CMOS
Hsiao [26]	3	-	-	Xilinx Virtex-4 FX100
Barranco [22]	4	640 × 480	32	Xilinx Virtex4 XC4vfx100
Blachut [27]	2	1280 × 720	50	Xilinx Virtex-7 VC707
This work	2	3840 × 2160	60	Xilinx UltraScale+ ZCU 104

**Table 2 sensors-22-05017-t002:** Hardware implementations of the HS algorithm on FPGA platform. Our solution can process a video in 4K resolution, but in multiple scales, which is rarely found in the literature and was done at most for Full HD resolution.

Implementation	Scales	Resolution	Iterations	FPS	Platform
Martin [30]	1	256 × 256	1	60	Altera APEX 20K
Bahar [31]	1	320 × 240	8	1029	Altera Cyclone II
Gultekin [32]	1	256 × 256	1	257	Altera Cyclone II EP2C70
Kunz [33]	1	640 × 512	30	30	Altera Stratix IV
Kunz [33]	1	4096 × 2304	20	30	Altera Stratix IV
Johnson [34]	1	3750 × 3750	10	30	Xilinx Virtex-7 VC707
Komor. [35]	1	1920 × 1080	32	60	Xilinx Virtex-7 VC707
Komor. [35]	1	1920 × 1080	128	84	Xilinx Virtex-7 VC707
Johnson [36]	1	1920 × 1080	15	200	Xilinx Virtex-7 VC707
Johnson [36]	1	3840 × 2160	15	48	Xilinx Virtex-7 VC707
Imamura [37]	2	1920 × 1080	32, 8	60	Custom 90nm CMOS
Bournias [38]	3	1024 × 1024	20, 10, 5	29	Altera Stratix V
This work	2	3840 × 2160	10, 5	60	Xilinx UltraScale+ ZCU 104

**Table 3 sensors-22-05017-t003:** Resource utilisation for the LK module in different data formats in 4K resolution. The resource usage decreases significantly as fewer pixels are processed in parallel. The 0.5ppc mode requires less memory elements, but a similar number of computing elements as the 1ppc mode. In general, many scales can be processed at the same time, but with reduced hardware utilisation.

Resource Type	4ppc	2ppc	1ppc	0.5ppc
LUT	47,683	26,254	15,574	15,565
Flip-Flop	82,482	49,987	33,999	33,981
Block RAM	92	50	25	17
DSP	540	290	165	165

**Table 4 sensors-22-05017-t004:** Resource utilisation for the HS module with 10 iterations in different data formats in 4K resolution. The resource usage decreases significantly as fewer pixels are processed in parallel. The 0.5ppc mode requires fewer memory elements, but a similar number of computing elements as the 1ppc mode. In general, many scales can be processed at the same time, but with reduced hardware utilisation.

Resource Type	4ppc	2ppc	1ppc	0.5ppc
LUT	43,265	21,907	11,462	11,408
Flip-Flop	61,530	31,405	16,958	17,069
Block RAM	134	69	39.5	26.5
DSP	488	244	122	122

**Table 5 sensors-22-05017-t005:** Comparison of *E_AAE_* errors in degrees for Middlebury dataset sequences. D—Dimetrodon, V—Venus, H—Hydrangea, G2—Grove2, G3—Grove3.

Implementation	Method	Version	D	V	H	G2	G3
Seyid [19]	Block	3 scales	8.23	6.41	14.80	5.80	10.90
Hsiao [26]	LK	1 scale	35.69	-	-	-	-
Hsiao [26]	LK	3 scales	21.35	-	-	-	-
Smets [29]	LK	2 scales	20.51	24.16	19.32	11.51	16.05
Smets [29]	LK	4 scales	10.15	16.21	8.28	5.50	10.08
This work	LK	1 scale	20.44	41.92	34.51	38.22	37.36
This work	LK	2 scales	12.54	28.14	18.08	17.81	24.88
Johnson [34]	HS	10 iter.	26.33	-	40.30	-	-
Johnson [34]	HS	50 iter.	21.32	-	36.93	-	-
Johnson [36]	HS	Precision	10.67	26.12	25.23	26.88	26.64
Johnson [36]	HS	Throughput	10.99	26.88	25.56	27.08	26.89
This work	HS	1 scale	32.27	41.99	35.61	33.11	35.68
This work	HS	2 scales	22.94	29.63	18.60	17.90	26.76

**Table 6 sensors-22-05017-t006:** Comparison of *E_AEE_* errors in pixels for Middlebury dataset sequences. D—Dimetrodon, V—Venus, H—Hydrangea, G2—Grove2, G3—Grove3.

Implementation	Method	Version	D	V	H	G2	G3
Seyid [19]	Block	3 scales	0.44	0.47	1.98	0.42	0.99
Hsiao [26]	LK	1 scale	2.16	-	-	-	-
Hsiao [26]	LK	3 scales	1.84	-	-	-	-
This work	LK	1 scale	1.02	3.33	2.47	2.22	3.11
This work	LK	2 scales	0.63	2.53	1.45	1.19	2.35
Johnson [34]	HS	10 iter.	1.18	-	2.71	-	-
Johnson [34]	HS	50 iter.	1.02	-	2.21	-	-
Johnson [36]	HS	Precision	0.63	2.34	2.23	1.56	2.53
Johnson [36]	HS	Throughput	0.65	2.43	2.34	1.66	2.62
This work	HS	1 scale	1.32	2.97	2.41	1.90	3.02
This work	HS	2 scales	1.01	2.31	1.40	1.21	2.38

**Table 7 sensors-22-05017-t007:** Resource utilisation for the LK algorithm on a ZCU 104 platform. Due to effective implementation of the multi-scale method, there is a small increase in resource utilisation (apart from BRAMs) when adding the second scale to the algorithm.

Resource Type	Available	Pass-Through	1-Scale Version	2-Scale Version
LUT	230,400	38,097 (17%)	89,167 (39%)	122,734 (53%)
Flip-Flop	460,800	44,673 (10%)	123,995 (27%)	183,688 (40%)
Block RAM	312	7 (2%)	119 (38%)	311 (100%)
DSP	1728	3 (0%)	559 (32%)	861 (50%)

**Table 8 sensors-22-05017-t008:** Resource utilisation for the HS algorithm on a ZCU 104 platform. Due to effective implementation of the multi-scale method, there is a small increase in resource utilisation (apart from BRAMs) when adding the second scale to the algorithm.

Resource Type	Available	Pass-Through	1-Scale Version	2-Scale Version
LUT	230,400	38,097 (17%)	84,477 (37%)	104,728 (45%)
Flip-Flop	460,800	44,673 (10%)	113,922 (25%)	145,872 (32%)
Block RAM	312	7 (2%)	161 (52%)	312 (100%)
DSP	1728	3 (0%)	507 (29%)	523 (30%)

## Data Availability

Not applicable.

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
