# Peer review of "Real-Time Efficient FPGA Implementation of the Multi-Scale Lucas-Kanade and Horn-Schunck Optical Flow Algorithms for a 4K Video Stream"

_sensors, 2022, doi:10.3390/s22135017_

Round 1

Reviewer 1 Report

The authors proposed an interesting work on an FPGA and compare it to similar approaches. Presenting an improved version of the optical flow algorithm.
Some points that might need to address by them are the following:

The authors compare at the introduction some methods for pixel displacement detection. Some complications are addressed with the Gabor filters and the Fourier transform. What is the benefit of the LK and HS in comparison with these other techniques?

After presenting Eq(1,2), what are x and y? I think that it should be stated that they correspond to the location in the vertical and horizontal axes of the image.

In line 168, is the value of neighboring pixels generally obtained by means of an average? Also, It would be convenient to explain which of both methods will be used in the article

In Eq(5), how does the parameter \alpha defined? Is it a weight? Also, do the \delta symbols imply differences between images in t and t+1? I suggest including equations in a more descriptive way.

In line 247. LUT is used for the first time. Use LUT (Look Up Table)

The authors should emphasize the importance of using an FPGA instead of a GPU. In lines 248 to 267, only a reduction in the fps was presented. No other tangible benefit was given.

It is odd that several subtitles of the article present the same name, i.e., Lucas-Kanade algorithm in sections 2.2, 3.1, 4.2.1. Same for the Horn-Schunck algorithm. A suggestion will be adding information to each one of the sections regarding the location. For example, Implementation in FPGA of the Horn-Schunck algorithm.

Author Response

Dear Reviewer,

We are very grateful for the time that you put into this review. We appreciate your comments and did our best to improve the paper. Most of the issues are addressed in the text and the changes are highlighted in blue colour. We also did multiple minor improvements to the language used in the paper. Please find below the responses to your particular remarks.

Reviewer #1:

The authors proposed an interesting work on an FPGA and compare it to similar approaches. Presenting an improved version of the optical flow algorithm.

Some points that might need to address by them are the following:

  1. The authors compare at the introduction some methods for pixel displacement detection. Some complications are addressed with the Gabor filters and the Fourier transform. What is the benefit of the LK and HS in comparison with these other techniques?

Thank you for this comment. In a general case, gradient-based algorithms are simple, easy to parallelise and yield dense optical flow. We added in Section 1 “Introduction”:

They are simple yet effective and relatively easy to parallelise, thus they can operate in real-time.

In terms of block methods, usually an interpolation is needed, when the matching is done for a specific set of points. It therefore increases latency and resource utilisation in case of a hardware implementation on an FPGA. We added in Section 1 “Introduction”:

However, to obtain dense flow like in gradient-based algorithms, an additional operation (potentially a complex interpolation) is needed.

In terms of more recent correlation methods, they can ensure good accuracy of the results, but they are computationally very demanding and usually not suitable for real-time processing on an FPGA. We added in Section 1 “Introduction”:

However, these methods are computationally demanding and not suitable for real-time operation.

Finally, the disadvantages of the neural network-based solutions were already addressed in the paper in Section 1 “Introduction”:

However, they cannot process very high-resolution video streams in real-time due to their complexity and number of parameters - best solutions (like LiteFlowNet2) can process up to 25-30 frames per second on Nvidia 1080 GTX GPU for the images of size 1024x436 pixels, which is nearly 20x smaller than a frame in 4K resolution. They also require a huge amount of training data and do not have a strong theoretical background as the “classical'' methods.

2. After presenting Eq(1,2), what are x and y? I think that it should be stated that they correspond to the location in the vertical and horizontal axes of the image.

Thank you for pointing it out. We should have definitely added this information before. Now it is added in Section 2 “The Horn-Schunck and Lucas-Kanade OF computation algorithms” below Eq. (1) and (2):

where x and y correspond to the horizontal and vertical location of the pixel in the image.

 3. In line 168, is the value of neighboring pixels generally obtained by means of an average? Also, It would be convenient to explain which of both methods will be used in the article

Thank you for this comment. The value of neighbouring pixels in local methods is usually calculated using the average in a window (sometimes weighted). We added this information and specified the types of the method realised in this work (HS - global, LK - local) in Section 2 “The Horn-Schunck and Lucas-Kanade OF computation algorithms”:

(...) while in the local methods the value of the determined flow is obtained only on the basis of the neighbouring pixels (usually by calculating their average). In this work, both approaches are presented in the subsections below and then implemented on a hardware platform - the HS algorithm (global method) and the LK algorithm (local method).

4. In Eq(5), how does the parameter \alpha defined? Is it a weight? Also, do the \delta symbols imply differences between images in t and t+1? I suggest including equations in a more descriptive way.

Thank you for your questions. The parameter α is defined as a constant numeric value, so it can be perceived as a weight between two factors. We added in Section 2.1 “Horn-Schunck algorithm”:

Its effect can be controlled by changing the constant numeric value of the parameter α.

The ∇ symbol has a meaning of a spatial gradient for each of the flow components to ensure their smoothness across the image. We added this information in Section 2.1 “Horn-Schunck algorithm”:

The other is responsible for the regularisation of the flow across the image, expressed as the sum of the squared magnitudes of the spatial gradients calculated on flow components u and v.

5. In line 247. LUT is used for the first time. Use LUT (Look Up Table)

Thank you for your comment. We added the abbreviation explanation in Section 3 “FPGA implementations of optical flow methods”:

The authors used temporal smoothing over 5 frames and a LUT (Look-Up Table)-based divider to reduce resource utilisation.

 6. The authors should emphasize the importance of using an FPGA instead of a GPU. In lines 248 to 267, only a reduction in the fps was presented. No other tangible benefit was given.

Thank you for your suggestion. In the mentioned part, we summarised the conclusions from one of the articles. We have already written some additional benefits of an FPGA over GPU there (Section 3 “FPGA implementations of optical flow methods”):

However, compared to GPUs, FPGAs are superior in several other aspects, which include the ability to operate without a computer as a host, smaller size and power consumption, and higher flexibility in programming.

We have also written some benefits of FPGAs in Section 4.1 “Video processing in 4K”, but we added more information:

FPGAs are much more energy-efficient, more flexible in programming and do not require a host computer like GPUs. They are also far more easy to use than a custom hardware (e.g. VLSI).

 7. It is odd that several subtitles of the article present the same name, i.e., Lucas-Kanade algorithm in sections 2.2, 3.1, 4.2.1. Same for the Horn-Schunck algorithm. A suggestion will be adding information to each one of the sections regarding the location. For example, Implementation in FPGA of the Horn-Schunck algorithm.

Thank you for your suggestion. We changed the titles of the subsections, so now there are no repetitive names. The following titles were modified:

Lucas-Kanade FPGA implementations (Section 3.1)

Horn-Schunck FPGA implementations (Section 3.2)

Implementation of the Lucas-Kanade algorithm (Section 4.2.1)

Implementation of the Horn-Schunck algorithm (Section 4.2.2)

Implementation of the multi-scale method (Section 4.3)

Reviewer 2 Report

In this work, the authors present a heterogeneous system for motion estimation using two algorithms. Overall, the paper is well written, with a very detailed related work section and providing useful insights to their key architectural desicions. The english language is very good without any  issues, making it an easy to read although a bit lengthy. The content is very good and all the necessary details are given.

My only remark is the comparisons are performed in a different architecture (as Table 2 illustrates) and thus we cannot safely evaluate whether the performance increase is due to the newer hardware or to a better architecture. I know that this is difficult, but the authors should provide a sufficient discussion about the extend of the improvement of the results due to the hardware.

Overall, I think this is a very good work and it is good the quality level of this journal.

Author Response

Dear Reviewer,

We are very grateful for the time that you put into this review. We appreciate your comments and did our best to improve the paper. Most of the issues are addressed in the text and the changes are highlighted in blue colour. We also did multiple minor improvements to the language used in the paper. Please find below the responses to your particular remarks.

Reviewer #2:

In this work, the authors present a heterogeneous system for motion estimation using two algorithms. Overall, the paper is well written, with a very detailed related work section and providing useful insights to their key architectural desicions. The english language is very good without any  issues, making it an easy to read although a bit lengthy. The content is very good and all the necessary details are given.

My only remark is the comparisons are performed in a different architecture (as Table 2 illustrates) and thus we cannot safely evaluate whether the performance increase is due to the newer hardware or to a better architecture. I know that this is difficult, but the authors should provide a sufficient discussion about the extend of the improvement of the results due to the hardware.

Overall, I think this is a very good work and it is good the quality level of this journal.

Thank you very much for your comments. A comparison with implementations on other platforms is a difficult issue indeed. Older solutions were implemented on hardware platforms that had insufficient amount of resources (computing, memory or both) for processing in 4K resolution. It was also not as popular back then as it is now. So some improvement is definitely due to using modern computing platforms. However, a better comparison may be done with recent solutions, realised on Xilinx VC707, Altera Stratix IV or Altera Stratix V platforms, which have similar amounts of resources. Our implementation is very effective due to the vector data format, which also changes between the scales. In this way, we achieved processing in 4K with multiple scales, which was not done before - in our opinion, this is the difference and improvement over other works in terms of the proposed architecture. We added a paragraph in Section 6 “Conclusions”:

Due to the hardware platform used - ZCU 104 with the Xilinx Zynq UltraScale+ MPSoC device, the algorithms implemented are not very complex. On the other hand, this modern device allows processing of a high-resolution data, which was not possible in case of older implementations (described in Sections 3.1 and 3.2) due to insufficient amount of computing resources or memory. However, compared with other modern solutions like works [32], [34], [35] or [37], where the authors used devices with similar amounts of resources, we applied a more efficient way of processing data by using the vector format. Furthermore, it was changing depending on the scale, thus allowing to significantly save computing resources in the multi-scale version of the proposed architecture.

Once again we are very grateful for your reviews and comments.

Sincerely,

Krzysztof Błachut

on behalf of the authors

Reviewer 3 Report

I understood that it was a study that proposed a method for processing 4K images in real time with FPGA.

Please answer the following items.

1.  What is the reason for processing 4K images with FPGAs instead of GPUs? Recently, many embedded GPUs such as small GPU (Orin) have been announced.

2. . In connection with, there is research on reconstructing low-resolution optical flow to high-resolution optical flow with neural networks. (4K + 50fps) Please state your advantages over these studies.

Author Response

Dear Reviewer,

We are very grateful for the time that you put into this review. We appreciate your comments and did our best to improve the paper. Most of the issues are addressed in the text and the changes are highlighted in blue colour. We also did multiple minor improvements to the language used in the paper. Please find below the responses to your particular remarks.

Reviewer #3:

I understood that it was a study that proposed a method for processing 4K images in real time with FPGA.

Please answer the following items.

  1. What is the reason for processing 4K images with FPGAs instead of GPUs? Recently, many embedded GPUs such as small GPU (Orin) have been announced.

Thank you for your question. FPGAs have numerous advantages over GPUs like smaller size, possibility to run without a host, lower energy consumption etc. In Section 3 “FPGA implementations of optical flow methods” we summarised one of the articles, in which the authors compared OF implementations on GPU and FPGA, with the latter much more energy-efficient. Compared with embedded GPUs, FPGAs may have similar possibilities, but they can still outperform eGPUs in specific tasks. In our opinion, one of them is the processing of a high-resolution video stream. Even if eGPU can also process such data, it still uses more energy than a SoC FPGA platform (10, 15 or 30 W power modes in Nvidia Jetson AGX Xavier vs 5-6 W for our optical flow implementation on ZCU 104). 

We do not have access to the Nvidia Orin yet, so it is difficult to compare it with the platform used in our implementation. It is definitely a “new generation” device, compared with our SoC FPGA, thus due to technological development its energy efficiency should be better than in the case of older devices, so it can be beneficial to implement and evaluate OF computation on it. On the other hand, for this type of device it is impossible to obtain the hardware architecture fully adapted to the implemented algorithm, which is the case for FPGAs.

  1. In connection with, there is research on reconstructing low-resolution optical flow to high-resolution optical flow with neural networks. (4K + 50fps) Please state your advantages over these studies.

Thank you for your comment. Unfortunately, we could not find the research mentioned, so we are not aware of details of this work and cannot compare it with our implementation. On one hand, we think that such a solution can be an efficient way of determining OF for high-resolution streams. On the other hand, some details (like small, thin objects) can be lost while calculating OF in low resolution. Another thing is that such a solution probably uses more energy than our implementation, unless it is implemented on an FPGA or using a VLSI custom architecture.

Once again we are very grateful for your reviews and comments.

Sincerely,

Krzysztof Błachut

on behalf of the authors

Round 2

Reviewer 3 Report

I'm sorry, I didn't show a detailed paper. https://www.mdpi.com/2073-8994/11/10/1251/htm The bibliography takes a similar approach. Why process 4K images with FPGA instead of GPU?

Author Response

Dear Reviewer,

Thanks for the information. We have provided an explanation below.

Reviewer #3:

  1. I'm sorry, I didn't show a detailed paper. https://www.mdpi.com/2073-8994/11/10/1251/htm The bibliography takes a similar approach. Why process 4K images with FPGA instead of GPU?

Thank you very much for the link. It is definitely a very interesting article, achieving state-of-the-art in this particular task (frame interpolation in 4K). It reconstructs optical flow in 4K resolution indeed, but some of the assumptions we made in the previous reply have proven true.

First of them is energy efficiency. According to Section 3.1 in the linked article, the authors used a Titan Xp GPU, which consumes around 250 W (without a host computer), according to documentation. Our implementation in SoC FPGA consumes below 6 W, so the difference is significant in favour of FPGA. If the aim is to e.g. use such an algorithm on the UAV, then FPGA is a good choice (eGPUs can also be used for this task, although they are not as effective as FPGAs and high-end GPUs).

The second concerns the time needed to compute one frame in 4K resolution. According to Table 3 in the mentioned article, at least 210 ms are required (and 380 ms without CUDA and cuDNN). This translates to 5 fps (and 3 fps, respectively). Our FPGA implementation runs at 60 fps, which is an order of magnitude faster. It can be discussed how much the computation of the optical flow and its rescaling to 4K takes, without any additional operations (the same for the power consumption), but there is no such information in the paper.

The third thing is that the optical flow is computed for the image of size 960x540 pixels and then it is upscaled. It can be discussed how many small details are lost during downscaling the image and not reconstructed correctly in 4K resolution (although the article shows promising results). Such an approach can also be implemented in an FPGA (but probably by using a simpler method of upscaling the flow) - then the architecture of the system can be simpler, using a vector format (4ppc) would not be necessary, while the entire system would require less hardware resources and less energy. However, our motivation was to be able to compute OF in 4K (and additionally in smaller resolutions) by proposing an effective way of data processing.

In conclusion, the aims of the two compared articles (from the link and ours) are different and cannot be directly compared. But our implementation in SoC FPGA runs at least 10x faster with significantly smaller energy consumption than the work done on this high-end GPU. Therefore, by using modern FPGA devices in particular tasks like parallel processing of a high-resolution video stream, we can achieve better parameters (real-time operation with very small energy consumption) than with the current GPUs.

However, the concept itself of calculating the flow in smaller resolution and then reconstructing it in 4K is very interesting, so we decided to add this paper to our bibliography and mention it in Section 1 “Introduction”:

Interesting alternatives can be found in the literature like [16], in which the authors calculate the flow using a neural network for 4K images reduced 4x in each dimension and then reconstruct it in the original resolution - however, with this approach there is a risk of losing some details.

Once again we are very grateful for the comments.

Sincerely,

Krzysztof Błachut

on behalf of the authors